# Mechanical and Electronic Properties of Al(111)/6H-SiC Interfaces: A DFT Study

**DOI:** 10.3390/molecules28114345

**Published:** 2023-05-25

**Authors:** Mostafa Fathalian, Eligiusz Postek, Tomasz Sadowski

**Affiliations:** 1Institute of Fundamental Technological Research, Polish Academy of Sciences, Pawińskiego 5B, 02-106 Warsaw, Poland; 2Department of Solid Mechanics, Lublin University of Technology, 20-618 Lublin, Poland

**Keywords:** DFT, interface, surface energy, young’s modulus, fracture toughness

## Abstract

A density functional theory (DFT) calculation is carried out in this work to investigate the effect of vacancies on the behavior of Al(111)/6H SiC composites. Generally, DFT simulations with appropriate interface models can be an acceptable alternative to experimental methods. We developed two modes for Al/SiC superlattices: C-terminated and Si-terminated interface configurations. C and Si vacancies reduce interfacial adhesion near the interface, while Al vacancies have little effect. Supercells are stretched vertically along the z-direction to obtain tensile strength. Stress–strain diagrams illustrate that the tensile properties of the composite can be improved by the presence of a vacancy, particularly on the SiC side, compared to a composite without a vacancy. Determining the interfacial fracture toughness plays a pivotal role in evaluating the resistance of materials to failure. The fracture toughness of Al/SiC is calculated using the first principal calculations in this paper. Young’s modulus (*E*) and surface energy (Ɣ) is calculated to obtain the fracture toughness (*K_IC_*). Young’s modulus is higher for C-terminated configurations than for Si-terminated configurations. Surface energy plays a dominant role in determining the fracture toughness process. Finally, to better understand the electronic properties of this system, the density of states (DOS) is calculated.

## 1. Introduction

Metal matrix composite materials (MMC) are made by combining a distinguished element or metal with another organic or ceramic material to achieve improved mechanical properties over conventional materials [1]. Metal matrix composites (MMCs) have properties due to their combination of two or more materials [2]. In today’s engineering world, aluminum metal matrix composites are used in applications such as aircraft, automotive, and aerospace components. Depending on the kinds of matrix and the reinforcement, there are different types of MMCs. It should be noted that most industrial MMCs include an aluminum matrix [3,4,5]. Aluminum matrix composites can lead to improved properties such as strength, stiffness, lowered density, electronic properties, and enhanced wear resistance which cannot be achieved with conventional material [1]. Many researchers tried reinforcing aluminum metal matrices with ceramic powder to increase their mechanical properties [6,7,8]. Adding high-strength particles to a ductile metal matrix produces a material whose mechanical and electronic properties are between the ceramic reinforcement and the base alloy. An aluminum metal matrix reinforced with silicon carbide particles is one of these types of alloy. Compared to other materials, matrix reinforced of Al and SiC has relatively high mechanical properties. Silicon carbide reinforcements are low-cost and improve yield strength, wear behavior, and elastic modulus without significantly compromising ductility [9]. Aluminum-based silicon carbide composites have been widely used in various parts of the industry due to their superior properties, which consist of desirable mechanical and electronic properties [10,11].

Compromises between computational and experimental results are a challenge for researchers. The experimental methods to investigate the overall behavior of Al/SiC composites are expensive, complex, and time-consuming to some extent. However, many simulation methods have been applied to studying Al/SiC particulate composites. Molecular dynamics (MD) and density functional theory (DFT) simulations are alternative approaches to investigating composites’ mechanical and electronic properties, such as Al/SiC. Recently, many studies have been conducted using the MD and DFT methods to analyze the mechanical, thermal, and electronic behavior of composites [12,13,14,15,16].

The position of the interface combination of composite materials can directly affect their performance. In particular, defects can have a noticeable impact on the interface of composition, and they can even directly affect the properties of the composite. Since studying the actual effect of a defect by the experimental method is costly, DFT calculations can be an appropriate choice [17]. This method allows us to conduct intense research by constructing just one defect. To elucidate the surface electronic states and properties of metal/ceramic interfaces, many DFT studies have been performed [18,19,20,21]. Xin-Yu Xu et al. [22] discussed the adhesion energy and electronic properties of the Al(111)/4H-SiC interface. Vacancies were considered by Chen et al. [23] as Au/MgO interface structures. Analysis of the effects of defects on the composite interface was carried out by Xue-ling Lin et al. [24]. In addition to studying the adhesion of metal/oxide interfaces, Matsunaka et al. also investigated the effects of oxygen vacancies on heterogeneous adhesions [25,26]. However, there have been few discussions of the deformation behavior of the interface and especially fracture toughness by DFT calculations. Fracture toughness, a key material property in materials science, refers to the critical stress intensity factor of a sharp crack at which the propagation of the crack suddenly becomes rapid and unlimited. The thickness of a component has a notable impact on the constraint conditions at the tip of a crack, with thin components typically exhibiting plane stress conditions and thick components experiencing plane strain conditions. The plane strain conditions, in particular, yield the lowest fracture toughness value, and this value is commonly referred to as the plane strain fracture toughness (*K_IC_*). The *K_IC_*, which is the critical value of the stress intensity factor observed under mode I loading and measured under plane strain conditions, is a vital parameter for characterizing a material’s fracture resistance [27].

Furthermore, Young’s modulus of the configuration of C-terminate and Si-terminate will be investigated. Finally, the density of state (DOS) plots will be performed for both positions to understand their electronic properties. Those are the topics we will examine in this paper.

## 2. Results and Discussion

### 2.1. Simulation Models and the Effects of Characteristic Defects

Vacancies in interfaces affect their mechanical and electronic properties, and their proportion is a key factor in modeling with density functional theory (DFT). The presence of vacancies changes stress distribution and local strain fields, influencing mechanical properties such as strength, ductility, and fracture toughness. It also creates localized states in the electronic band structure, altering electronic properties such as conductivity and optical properties. Defect concentration, nucleation, propagation of dislocations, deformation mechanisms, and density and distribution of localized states in the electronic structure depend on the proportion of vacancies [28]. This study focuses on the investigation of the impact of a vacancy at the interface in distinct positions. To investigate the characteristic of Al/SiC, the first geometrical parameters of this structure were designed and optimized under DFT framework calculations, Figure 1 and Figure 2. Eight layers of Al(111) are placed on eleven layers of SiC. The Al(111)/6H-SiC supercell models of 5×5 were constructed to study the effects of point vacancies at the interface, and a single layer of hydrogen atoms was added to the bottom of SiC. In fact, H atoms are introduced to remove the asymmetry-induced magnetization of 6H-SiC along the z-direction. According to the feasibility of this simulation, the vacuum thickness is considered to be about 25 Å. It can be seen that the Al-Si bond length in the most equilibrium condition was obtained at about 2.50 Å, and this value was 1.95 Å in Al-C which is in close agreement with previous relevant research results by B. K. Rao [29].

The interface between materials has a direct and significant impact on the properties of composite materials. This is due to the fact that the interface region is the zone where the different components of the composite material interact and influence each. The Al(111)/6H-SiC(0001) interface can be investigated using DFT calculations to gain insights into interfacial bonding, structural properties, electronic properties, and surface termination. Determining the interfacial bonding between Al and SiC is crucial for understanding the interface’s strength and stability. Investigating the structural properties, such as the interfacial distance and lattice mismatch, can provide important information about the interface’s stability and electronic properties. DFT calculations can also predict the electronic band structure, density of states, and impact of defects and surface termination on the electronic properties. Studying the impact of surface termination on the properties of the interface can help identify the differences in electronic properties of different surface terminations.

In the case of the Al(111)/6H-SiC interface, there are two configurations of C-terminated (Figure 1) and Si-terminated (Figure 2) at the interface. Due to the distinct categorization of atoms at the interface of 6H-SiC (0001), it is imperative to consider both the C-terminated interface and the Si-terminated interface. For each configuration, seven models are structured, including six different point vacancies and a supercell defect-free, as contrasted in Figure 3 and Figure 4. For example, to begin with, one aluminum atom is removed at the interface of the Al side, and this structure model is named Al_1_. After that, a single point vacancy is created from the second layer at the Al interface called Al_2_. Then, based on this procedure, four other models are generated in the case of C_1_, C_2_, Si_1,_ and Si_2_, representing different point vacancy models near the interface. Therefore, the mechanical properties of Al/SiC, including the interface with defects and defect free, are investigated.

### 2.2. Formation Energy and Work of Adhesion Energy

The formation energy of a single vacancy at the interface can be calculated by Equation (1):(1)$$E_{f}=(E_{v}+X_{v}-E)/A,$$
where $E_{v}$ is the free energy (eV) of the system with vacancy, *E* is the free energy (eV) of the corresponding complete supercell without vacancy, $X_{v}$ is the potential energy (eV) of the original atom in the vacancy, and *A* is the total surface area of the interface (${Å}^{2}$). Table 1 shows the formation energy for each vacancy position and defect-free system.

The work of adhesion is given by Equation (2) [30,31,32]:(2)$$W_{ad}=(E_{Al}+E_{SiC}-E_{Com})/A,$$
where $E_{Al}$
${and\;E}_{SiC}$ are the total energy of the relaxed Al(111) and 6H-SiC slabs, respectively. Moreover, $E_{Com}$ is the total energy of the Al(111)/6H-SiC composite after optimization. Adhesion energy for each of the vacancy positions and the defect-free is shown in Figure 5.

C and Si vacancies’ formation energy is much higher than Al point vacancies energy. It can mean that Al vacancies are formed more easily than C and Si vacancies. According to Finnis’ research [33], *W_ad_* can directly reflect wettability. Therefore, we can compare interfacial wettability by studying interfacial adhesion energy. The $W_{ad}$ data shows that Al vacancies have approximately no impact on the surface improvement for C-terminated and Si-terminated configurations. It should be noted that the adhesion energy at Al with a defect at the interface is lower than the ideal configuration without vacancy. Considering the interfacial Al vacancy, the adhesion energy at the interface is lower than in the ideal configuration without the vacancy. Hence, the wettability of the interface is weakened instead of improved. A substantial reduction in the adhesion energy is associated with the C and Si vacancies, which correlates with the interface’s wettability. Vacancies have caused electron rearrangements at the interface, and most rearrangements do not improve the interface’s performance. The formation energies (*E_f_*) for vacancies in carbon (C) and silicon (Si) are very close to each other, and they are markedly higher compared to vacancies in aluminum (Al). It can be seen in C-terminated and Si- terminated configurations the lowest *E_f_* appears at the interface on the SiC side. Even the most minor formation energy of Si or C is higher than that of Al vacancies. Despite this, the minor formation energy does not appear at the surface layer (Al_1_) but at the layer’s position (Al_2_). This is likely due to the strong bonding between SiC and the surface Al atoms, which causes the Al_1_ electrons to move toward the SiC and weakens the effect on the Al subsurface. It is evident that for Al(111)/6H-SiC interface, a feasible vacancy is an Al_2_ vacancy, and a vacancy can hardly occur in SiC. Therefore, it is challenging to create a vacancy on the SiC. In contrast, on the Al side, the vacancies are much easier to form, and the surface vacancies (Al_1_) decrease the wettability between SiC and Al.

Fundamentally, for metal layers, the presence of vacancies in Al layers does not affect the density of states of free electrons [33]. Therefore, if we consider the Al/SiC analysis as a whole, a more significant interference can be caused by the free charge of Al near the Fermi level. Thus, we only consider the electron density change on the SiC side in the composites’ d = Density of State (DOS) analysis.

### 2.3. Young’s Modulus

We also calculated Young’s modulus of the Al/SiC. For this purpose, all these composite configurations were compressed and then stretched along the z direction with a small increment (1.00709 Å) [34,35,36]. Finally, the strain energy for each strain value was plotted as shown in Figure 6 and Figure 7. Generally, Young’s modulus can be calculated as the second derivative of the total energy of the systems over the equilibrium volume, where ε is the strain as illustrated in the following formula [37]:(3)$${E=\frac{1}{V}\left(\frac{d^{2}U}{d\varepsilon^{2}}\right)}_{\varepsilon =0}$$
where *V* is defined as the volume of the Al/SiC supercell, *U* is potential energy, and *ε =* Δ*L/L*_0_, in which Δ*L* is changed in bond length concerning the initial bond length. It should be noted that these total energy values were fit to a polynomial. We calculated Young’s modulus of SiC/Al according to Equation (3), Figure 8. After calculating Young’s modulus, the diagram shows that Young’s modulus in the C-terminated configuration is higher than Young’s modulus in the Si-terminated configuration. In addition, the highest value of Young’s modulus consists of defect-free and Al_2_, and its lowest value is related to C_1_ in the C and Si termination. The results show that while the defects on aluminum layers do not cause visible changes in the composite’s modulus of elasticity, the presence of vacancies on the SiC side deteriorates the elastic modulus, especially when the vacancies happen in the carbon atoms. It is noteworthy that Young’s modulus of pure silicon carbide is much higher than Young’s modulus of pure aluminum. Therefore, Young’s modulus significantly decreases when a vacancy occurs on the SiC side.

### 2.4. Investigation of Tensile Test Simulation Behavior

To study Al(111)/6H-SiC composite, the tensile test is simulated to investigate the tensile fracture processes. For tensile simulation, all atomic positions were gradually displaced in the *z*-direction with small increments (1.00709 Å). The atoms of the whole system maintain the relative position along the *z*-direction concerning the new configuration. Then, the atomic positions of the new configuration are entirely relaxed at each displacement step. In other words, the stress–strain curves were obtained by gradually deforming the modelled cell in the direction of the applied displacement and simultaneously relaxing both the atomic basis vectors orthogonal to the applied displacement. As the SiC/Al structure is an atomic configuration in the DFT calculation, the stress value of each graph represents the average stress of the atoms [38,39].

The stress tensor *σ* is defined in terms of the individual strain tensor components *ε_ij_* by Equation (4):(4)$$\sigma =\frac{1}{Ω}\left(\frac{\partial U}{{\partial \varepsilon }_{ij}}\right),$$
where *U* is the total energy and Ω the supercell volume [40]. The tensile stress–strain relation includes two main models in Figure 9: C-terminated and Si-terminated. Additionally, we depicted the stress–strain curves in seven different situations, including defect-free, vacancy Al_1_, Al_2_, and vacancy Si_1,_ Si_2_, C_1,_ and C_2_.

The presence of defects is an essential factor that directly affects the ultimate tensile strength. Vacancies in the layers near the interface can impact the total energy. The value of the ultimate strength of each model is shown in Figure 9. For the C-terminated configuration, Figure 9a, the ultimate strength value belongs to the Si_2_ model (5.8 GPa). In comparison, the lowest one is related to Al/SiC without defect (3.8 GPa). When silicon carbide (with very high stiffness) is added to aluminum (with low stiffness), a high-stress concentration is created in the interface area. While creating vacancies in the SiC region reduces its stiffness and the stress concentration at the interface; as a result, we will have better tensile properties despite the defects on the SiC side. The interface binding ability of SiC is enhanced to varying degrees by vacancies of C and Si. Therefore, the tensile properties in the second layer of vacancies close to the interface on the SiC side are somewhat improved. On the aluminum side, the elongation of Al_2_ is slightly reduced compared to the perfect composite, while in Al_1,_ the ultimate strength and elongation go up slowly to 5.3 GPa and 0.11, respectively. It may be due to the strong bond between SiC and Al atoms on the interface, as well as higher adhesion energy in Al_1_.

According to the Si-terminated configuration, Figure 9b, the composite with Si_1_ vacancies has a high tensile strength of about 5.1 GPa, and its elongation is over 0.11. Compared to the perfect composite (4.4 GPa), the tensile strength is increased steeply; however, the elongation is similar. On the SiC side, C and Si vacancies also partially improve the bonding ability of the interface. As observed, the Si vacancy at the second layer (Si_2_) significantly impacts the bonding performance. It can be due to the large size of the Si atom. When a Si_2_ atom is removed, a large vacancy is created. The vacancy can bring some lattice distortion near the point, and it eventually will affect the properties of the materials to some extent. The second-highest ultimate tensile strength curve belonged to Si_1_(4.9 GPa), but it has the highest elongation (over 0.12). On the aluminum side, the tensile strength at Al_1_(4.51 GPa) (vacancy at the interface) is higher than the tensile strength without a vacancy (4.3 GPa). When penetrating the second layer (Al_2_), Al vacancy significantly affects the tensile properties. During the tension process, the vacancy effect at the second layer (Al_2_) is similar to the situation without vacancy.

In addition, we found that the failure positions do not appear at the interfaces of the models when they are stretched. According to this, the adhesion energy of the interface does not influence the material’s brittleness. The weakest binding layers determine the material’s tensile fracture properties. For Al(111)/6H-SiC(0001) interface, the weakest position occurs between the first and the second layer at the Al side rather than the interface. The greatest elongation is found for the Si_2_ model in C and Si-terminated configurations. In tensile testing of Al/SiC composite materials, fractures are observed to occur in the first or second layer of Al rather than at the interface between SiC and Al. This behavior is due to the weaker bonding between the Al atoms and the SiC substrate in the first few layers on the Al side. During tensile testing, stress is first applied to the weakest binding layers of the composite material. In the case of the Al/SiC composite, the first few layers on the Al side have weaker bonding with the SiC substrate compared to the interface between SiC and Al, making these layers more susceptible to deformation and failure. At the interface between SiC and Al, there are strong covalent bonds between the carbon atoms in SiC and the aluminum atoms in Al. However, in the first few layers of Al, the bonding strength between the Al atoms and the SiC substrate is weaker, as it involves weaker Van der Waals forces and weaker metallic bonds [41,42]

### 2.5. Surface Energy and Fracture Toughness

To study the Al/SiC interfacial bonding, we should first investigate the surface energy of these two materials. It is expressed by Equation (5) [43,44,45]:(5)$${Ɣ}_{surf}=\frac{E_{slab}-(\frac{N_{slab}}{N_{bulk}})E_{bulk}}{2\times A_{slab}}$$
where $E_{slab}$ is the total energy of the system containing the vacuum layer, $E_{bulk}$ is bulk energy per atom, *N_slab_* is the total number of atoms in the slab structure, and *A_slab_* is the area of the surface unit cell. Generally, hexagonal 6H-SiC has an indexed surface (0001) with two different C and Si-termination surface statuses. Moreover, it illustrates that the stability of 6H-SiC is much lower than the Al index surface; therefore, it severely tends to adsorb Al atoms to decrease the force imbalance.

Regarding materials science, fracture toughness is the ability of a material containing a crack to resist fracture. The fracture can be defined as the separation of a heterogeneous interface into two distinct homogeneous components in a bonded interface. A material’s fracture toughness *K_IC_* can be calculated using the following equation as a critical stress intensity factor [46,47,48]:(6)$$K_{IC}=\sqrt{4ƔE}$$
where Ɣ and *E* are surface energy and Young’s modulus, respectively. Their values are obtained from the DFT calculations.

Fracture toughness is calculated for C-terminated and Si-terminated configurations. Additionally, fracture toughness is obtained for different vacancies, Figure 10. Table 2 provides information about the surface energy and fracture toughness of the Al/Si composite.

According to Table 2 and Figure 10, the values of fracture toughness for C-terminated are greater than Si-terminated to some extent. Additionally, the C-termination interface has the highest surface energy. Therefore, in the C-terminated configuration, the perfect composite has the highest fracture toughness (1.40 MPa m^1/2^). In contrast, the smallest fracture toughness belonged to C_2_ (0.85 MPa m^1/2^). The fracture toughness of the defect-free composite and Al_1_ is of minor difference. However, there is a slight difference between the fracture toughness of Al_1_ and Al_2_. It can be due to the larger surface energy of Al_1_ compared to Al_2_ on the interface. In other words, the deeper the defects on the aluminum side, the more toughness can be reduced. On the other hand, the fracture toughness of the Al/SiC composite can be dramatically affected by C and Si vacancies. It can be attributed to the surface energy reduction at the interfacial area.

In the Si-terminated configuration, fracture toughness decreases particularly on the perfect composite and Al sides (1.21, 1.21, and 1.20 MPa m^1/2^, respectively). Subsequently, there is a noticeable reduction in the surface energy for the composite without defect, Al_1_, and Al_2_. However, there are no impressive changes in fracture toughness or surface energy on the SiC side. Since *K_IC_* is experimentally measured under polycrystalline conditions, SiC and Al volume fractions significantly affect the composites’ fracture toughness. Effects are also associated with grain boundaries. Hence, combining these points can make DFT calculation for the *K_IC_* of the Al/SiC interface reasonable and acceptable.

### 2.6. Density of States (DOS)

We also investigated the density of states (DOS) illustrated in Figure 11 to understand these systems’ electronic properties better. According to the calculated DOS diagrams for C-terminated configuration Figure 11a, the presence of Al defects has little effect on the DOS distribution of SiC. Due to the presence of Al vacancy defects near the interface, the effects of Al on SiC are reduced, which manifests as reduced DOS near the Fermi surface. Due to C vacancies near the interface, the charge distribution in the conduction band is enhanced, and the valence band shifts to the left, reducing energy. It happens of the strong ability of C atoms to gain electrons, which causes the Si atoms to transition external electrons to C. Absent the C atom, the Si atom loses part of its external electron bond. The graph illustrates that the presence of Si vacancies near the interface, particularly the sublayer Si vacancies, causes an increase in the density of states near the interface to some extent.

Si-terminated configurations are analogous to C-terminated ones, as shown in Figure 11b; however, the impact of Al, C, and Si vacancies is slightly weaker. Due to the Si-terminated interface’s lower adhesion energy than the C-terminated configuration, the total surface charge distribution is less affected by C, Si, and Al vacancies. There is a difference in that the sublevel Si vacancies approximately do not influence the DOS of SiC. Hence, it can be further confirmed that, for the SiC side of the Al/6H-SiC interface, the effect of the vacancies on the material is concentrated near the interface.

Based on the analysis, it is evident that SiC exhibits metallic characteristics due to the band structure of SiC across Fermi levels when combined with Al in both superlattices. However, the Si-terminated interface displays a less dense band structure across Fermi levels as compared to the C-terminated interface. This indicates that the C-terminated interface has stronger interfacial adhesion and bond strength. The figures presented in the analysis suggest that the Fermi energy levels inside SiC are largely unoccupied, and the electronic states below the Fermi energy level are primarily bonding states that remain stable. For Si_2_, the density of the state near Fermi levels is high. For Si_1_, the density of the state near Fermi levels is lower than Si_2_, which means much more stable than Si_1_. After conducting an analysis, it has been observed that there is a difference in the density of states near Fermi levels between Si_1_ and Si_2_. Specifically, Si_2_ has a higher density of states near Fermi levels compared to Si_1_. This indicates that Si_2_ is less stable than Si_1_. On the other hand, Si_1_ has a lower density of states near Fermi levels, suggesting that it is more stable than Si_2_.

Upon analysis, it has been observed that Al_1_ and Al_2_ display a low state density, indicating a tight connection and structural stability between Al and C. Of the two, Al_2_ displays the highest state density, implying an unstable position located at the second layer from the interface of the Al slab.

## 3. Methodology

In this work, the atomic geometry and electronic structure of Al(111)/6H-SiC are calculated by the density functional theory (DFT) framework [49,50] and executed using the Spanish Initiative for Electronic Simulations with Thousands of Atoms (SIESTA) code [51,52,53]. We used the generalized-gradient approximation (GGA) function with the Perdew–Burke–Ernzerhof (PBE) [54,55] to treat the effects of correlation and electronic exchange. All atomic orbital basis sets were double-ξ plus polarization orbitals (DPZ) with a 50MeV energy shift and 0.3. A split norm [5 × 5 × 1] Monkhorst–Pack grid [56,57] was used for the k-point sampling of the Brillouin zone, and the atomic locations were relaxed until the remaining forces on any atom were smaller than 0.02 eV${Å}^{-1}$ [58,59,60]. The cutoff of the plane-wave kinetic energy is 120 Ry in the calculations. The study employs the following cell parameters for its calculations: a = 15.406 Å, b = 13.342 Å, c = 41.993 Å, and α = 90°, β = 90°, γ = 120°. The ground state of the electrons can be found by solving the Kohn–Sham equation. Periodic boundary conditions were used with 5 × 5 supercells. The vacuum height was set to 25 Å to ensure that the *z*-axis of the periodic supercell is large enough and to eliminate spurious interactions between SiC/Al images periodically repeated. The generated samples are all fully relaxed in three directions before performing stress–strain calculations.

To calculate Young’s modulus, the Al/SiC configurations (C- and Si-terminated), were compressed and then elongated along the *Z*-axis with an increment of 1.00709 Å, and as a result, the strain energy–strain curve was plotted. Then, we can calculate Young’s modulus using the derivation of the equation previously mentioned, which is the second derivative of the system over equilibrium volume. Moreover, uniaxial tension is obtained by holding the longitudinal strain fixed in the *Z*-axis direction while relaxing the structures. Meanwhile, the supercell remains fixed in the *X*-axis and *Y*-axis directions. Finally, the fracture toughness *K_IC_* of a material is calculated from Equation (5) since this property is an inherent property of the material where the surface energy Ɣ and Young’s modulus *E* are obtained from DFT calculations described previously.

## 4. Conclusions

To summarize, we investigated the effect of different vacancies on the interfacial behavior of SiC/Al composites under C-termination and Si-termination two interface configurations by DFT calculations to analyze the surface stability, mechanical behavior, and fracture mechanism of 6H-SiC/Al(111) interfaces. In addition, the nature of bonding, work of adhesion (*W_ad_*), surface energy (*γ*), and fracture toughness (*K_IC_*) at the interfaces are investigated. The main conclusions are as follows:The vacancies significantly impact interface adhesion. Most vacancies weaken interfacial adhesion, though the weakening effects reduce with the depth of the vacancy from the interface to the inner.Although Al/SiC composites have lower adhesion energy due to defects at the interface, their tensile strength increases. It is because combining Al with SiC produces a much stronger compound than pure.The presence of Al vacancies at the interface may somewhat raise the material’s tensile properties by restoring Al atoms at the interface. According to the tensile strength results, the tensile properties of the configurations are better when point vacancies are placed on the SiC side than the point vacancies could be placed on the Al side.Vacancies on the SiC side significantly affect Young’s modulus, especially in the C_1_ model. Furthermore, Si-terminated configurations have a significantly lower Young’s modulus than C-terminated configurations. It is notable that Young’s modulus decreases significantly in the presence of carbon vacancies at the interface.C and Si vacancies affect fracture toughness more than Al vacancies in the interface. In the perfect model, the fracture toughness is 1.40 MPa m^1/2^ and 1.21 MPa m^1/2^, respectively, for the C-terminated and Si-terminated configurations. The lowest fracture toughness was found in the C_1_ model with C- and Si-terminated configurations (0.85 MPa m^1/2^ and 0.83 MPa m^1/2^, respectively). Likewise, the lowest Young’s modulus was observed in C_1_ for C- and Si-terminated configurations (145.1 GPa and 138.5 GPa, respectively).

Experimenting with Al/SiC composites at the atomic level can be challenging. As a result, DFT calculations can be an economical method for investigating the effect of vacancies on composite materials. The subsequent research will investigate different ceramic materials and compare the DFT and experimental results.

## Figures and Tables

**Figure 1 molecules-28-04345-f001:**
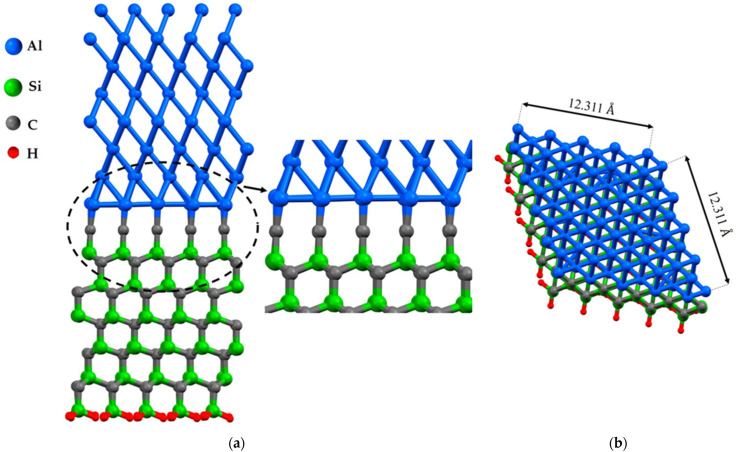
The structures of Al(111)/6H-SiC interfaces of the C-terminated configuration without defect after relaxation: (**a**) the side view; (**b**) the top view.

**Figure 2 molecules-28-04345-f002:**
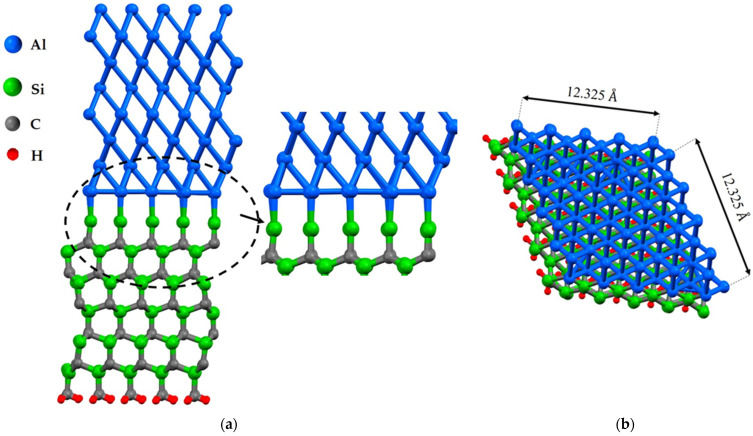
The structures of Al(111)/6H-SiC interfaces of the Si-terminated configuration without defect after relaxation: (**a**) the front view; (**b**) the top view.

**Figure 3 molecules-28-04345-f003:**
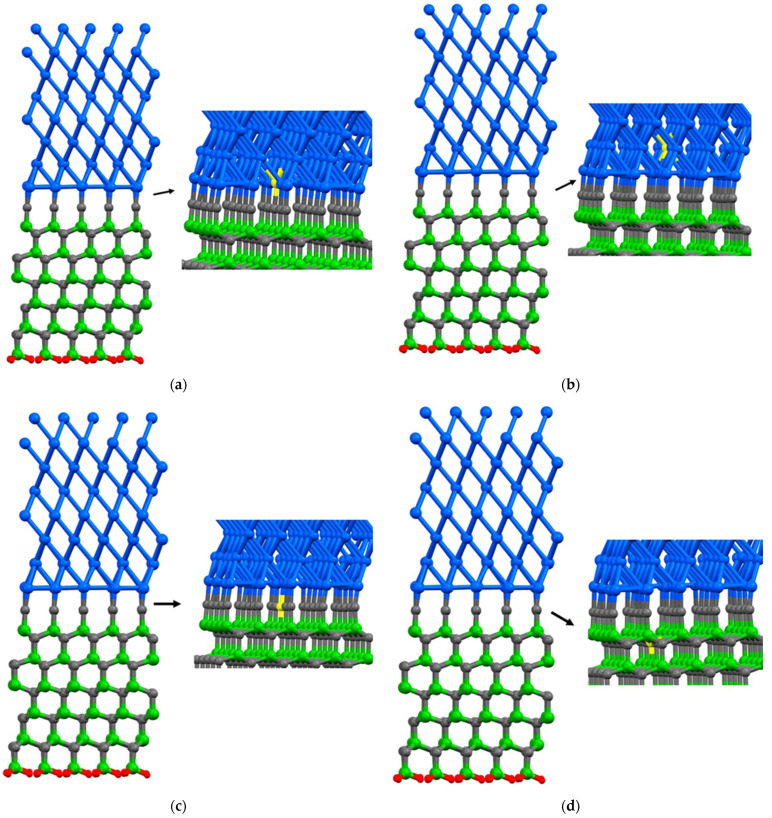
A single point vacancy is removed from the corresponding layer in the C-terminated interface configuration: (**a**) Al_1_; (**b**) Al_2_; (**c**) C_1_; (**d**) C_2_; (**e**) Si_1_; (**f**) Si_2_. The colors blue, green, gray, and red are representative of aluminum, silicon, carbon, and hydrogen atoms, respectively. Additionally, the color yellow is employed to signify the absence of an atom, as it represents a vacancy within the atomic structure.

**Figure 4 molecules-28-04345-f004:**
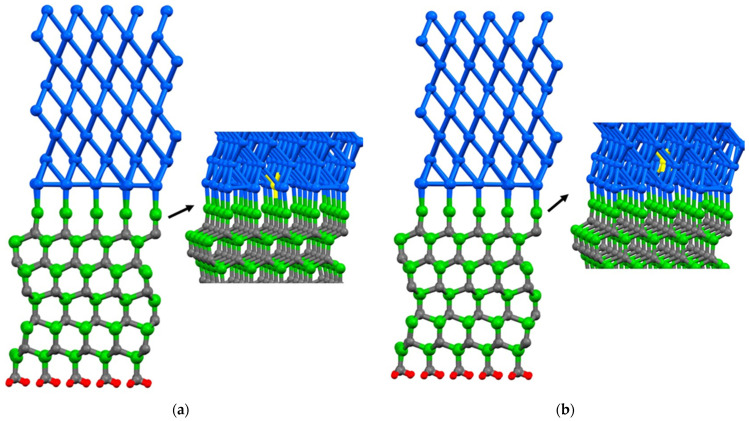
A single point vacancy is removed from the corresponding layer in the Si terminated interface configuration: (**a**) Al_1_; (**b**) Al_2_; (**c**) Si_1_; (**d**) Si_2_; (**e**) C_1_; (**f**) C_2_. The colors blue, green, gray, and red are representative of aluminum, silicon, carbon, and hydrogen atoms, respectively. Additionally, the color yellow is employed to signify the absence of an atom, as it represents a vacancy within the atomic structure.

**Figure 5 molecules-28-04345-f005:**
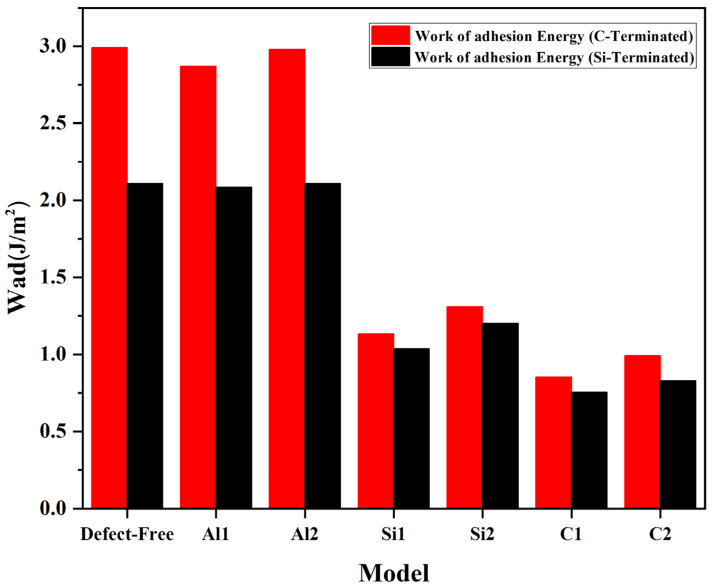
The adhesion energy of Al/SiC interfaces with different point vacancies for C-terminated configuration and Si-terminated configuration.

**Figure 6 molecules-28-04345-f006:**
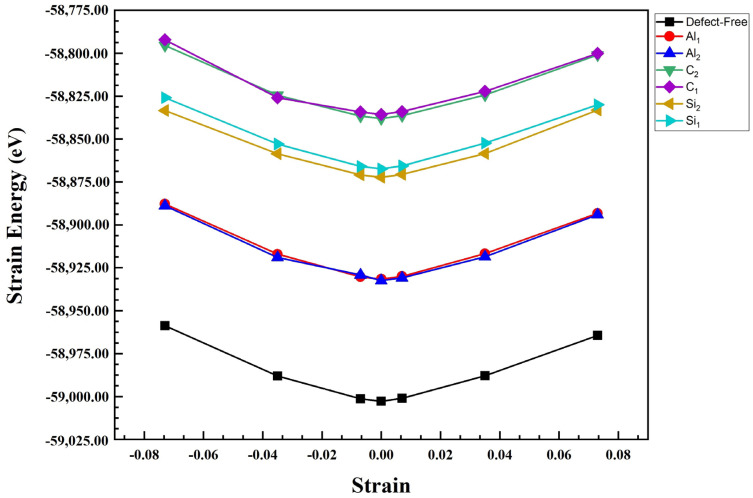
Strain energy versus strain for uniaxial strain in Al_1_, Al_2_, C_1_, C_2_, Si_1_, Si_2_, and defect-free of Al/SiC interfaces for C-terminated configuration.

**Figure 7 molecules-28-04345-f007:**
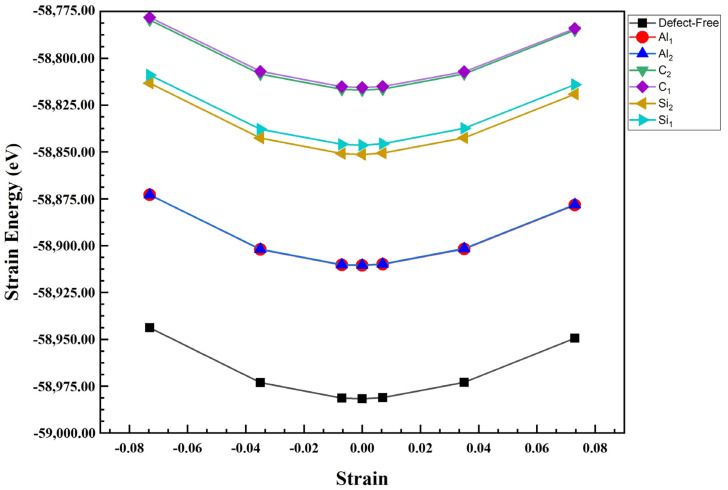
Strain energy versus strain for uniaxial strain in Al_1_, Al_2_, C_1_, C_2_, Si_1_, Si_2_, and defect-free of Al/SiC interfaces for Si-terminated configuration.

**Figure 8 molecules-28-04345-f008:**
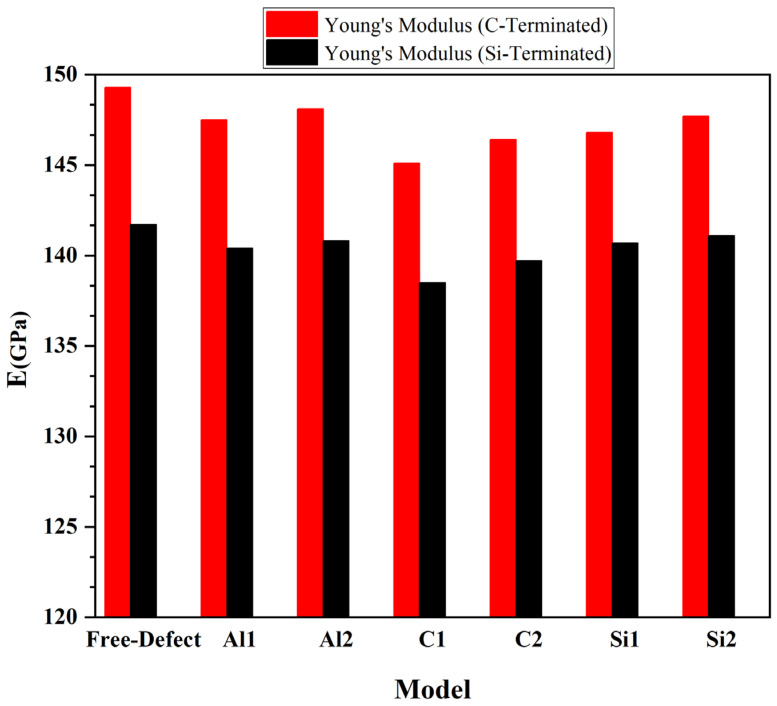
Young’s modulus of SiC/Al for C-terminated and Si-terminated interface configurations.

**Figure 9 molecules-28-04345-f009:**
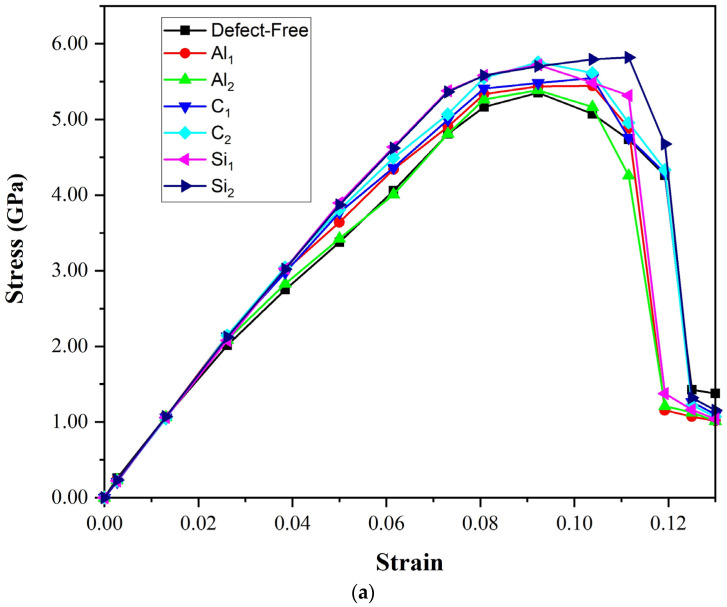
The strain–stress relation of the (**a**) C-terminated and (**b**) Si- terminated Al/SiC interfaces with different vacancies.

**Figure 10 molecules-28-04345-f010:**
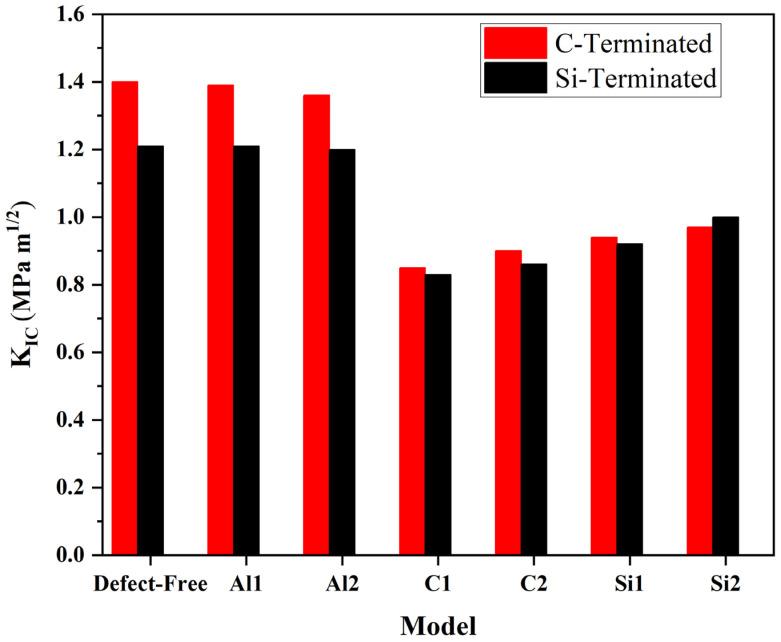
Fracture toughness for C- and Si-terminated configurations.

**Figure 11 molecules-28-04345-f011:**
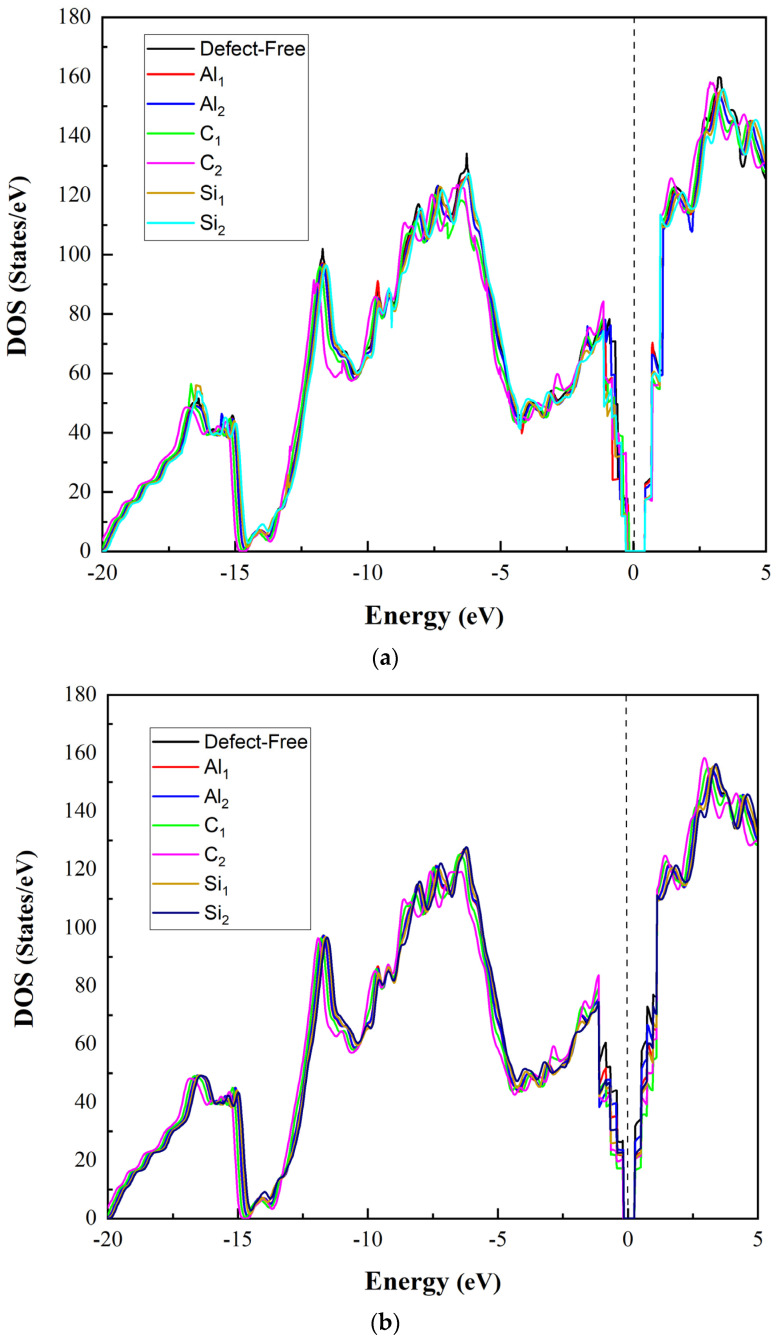
The density of state (DOS) of the Al/SiC interface: (**a**) C-terminated configuration; (**b**) Si-terminated configuration.

**Table 1 molecules-28-04345-t001:** The interfacial adhesion energy and formation energy of vacancies.

	Model	$W_{ad}$ (J/m^2^)	*E_f_* (J/m^2^)	*d* (Å)
C-Terminated	No Vacancy	2.991	-	1.95
	Al_1_	2.870	1.740	1.92
	Al_2_	2.981	1.511	1.93
	Si_1_	1.132	2.212	1.98
	Si_2_	1.200	2.592	1.98
	C_1_	0.853	2.641	2.01
	C_2_	0.992	2.977	2.00
Si-Terminated	No Vacancy	2.109	-	2.42
	Al_1_	2.085	1.702	2.52
	Al_2_	2.108	1.518	2.53
	Si_1_	1.036	2.161	2.54
	Si_2_	1.310	2.407	2.53
	C_1_	0.753	2.649	2.53
	C_2_	0.828	2.956	2.53

**Table 2 molecules-28-04345-t002:** The calculated surface energy and fracture toughness for Al/SiC.

	Model	Ɣ (J/m^2^)	*K_IC_* (MPa m^1/2^)
	No Vacancy	3.29	1.40
	Al_1_	3.28	1.39
	Al_2_	3.17	1.36
C-Terminated	Si_1_	1.53	0.94
	Si_2_	1.60	0.97
	C_1_	1.25	0.85
	C_2_	1.39	0.90
	No Vacancy	2.60	1.21
	Al_1_	2.60	1.21
	Al_2_	2.58	1.20
Si- Terminated	Si_1_	1.53	0.92
	Si_2_	1.81	1.00
	C_1_	1.25	0.83
	C_2_	1.32	0.86

## Data Availability

The data is not available due to ongoing project.

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
