# Peer review of "Mechanical and Electronic Properties of Al(111)/6H-SiC Interfaces: A DFT Study"

_molecules, 2023, doi:10.3390/molecules28114345_

Round 1
Reviewer 1 Report
- Please make it clear at the beginning of the article that the article is about the interface between the Al (111) surface and the 6H-SiC (0001) surface.
- Did the authors perform convergence tests on the number of surface layers of Al (111) and 6H-SiC? Is a model consisting of 11 layers of SiC and 8 layers of Al sufficient to produce accurate results?
- Authors should provide evidence or reference to substantiate their statement “Fundamentally, for metal layers, the presence of vacancies in Al layers does not affect the density of states of free electrons.” on page 6, line 138.
- Part of page 8, line 153, is obscured.
- On page 12, line 197, the ultimate strength without defects should not be 3.8 Gpa.
- On page 12, line 206, how did the authors determine that the increase in strength and elongation of Al1 is due to the strong interaction between SiC and Al? And the adhesion energy of Al1 should be lower.
- Authors should provide evidence and a detailed description to substantiate their statement “In addition, we found that the failure positions do not appear at the interfaces of the models when they are stretched. According to this, the adhesion energy of the interface does not influence the material's brittleness. The weakest binding layers determine the material's tensile fracture properties. For Al (111)/6H-SiC (0001) interface, the weakest position occurs between the first and the second layer at the Al side rather than the interface.” on page 13, line 222.
- Authors should describe the significance of the numerical magnitude of the KIC.
- Authors should provide a more detailed description of the DOS diagram or redraw the DOS diagram, which at this stage is difficult to understand with the DOS diagram and analysis.
Author Response
Reviewer 1
Dear Reviewer,
Thank you for your valuable comments. We prepared the following explanations.
Comment 1:
Please make it clear at the beginning of the article that the article is about the interface between the Al (111) surface and the 6H-SiC (0001) surface.
We added the explanations in the text (yellow lines) [Results and Discussion, lines: 110-122 and 124-126 in the Word manuscript file]
Comment 2:
Did the authors perform convergence tests on the number of surface layers of Al (111) and 6H-SiC? Is a model consisting of 11 layers of SiC and 8 layers of Al sufficient to produce accurate results?
Yes, in the initial phase, Al and SiC were subjected to individual relaxation procedures. Subsequently, optimization of Al(111)/6H-SiC was conducted by varying orientations and different initial parameters such as pseudopotentials, basis set, k-point grid, and cutoffs. Finally, the most stable configuration was selected as the definitive structure for both C and Si-termination configurations.
Indeed, we are of the opinion that the number of layers and atoms employed in our study suffices for our objectives. It is noteworthy that several published articles have explored the mechanical and electronic properties of interface areas using DFT calculations with a relatively small number of atoms and layers [1-3]. Hence, we believe that the chosen number of layers and atoms in our study is adequate for a meaningful analysis of the system under consideration. At the end of the paragraph, we have appended several references for further reading.
Density functional theory (DFT) can provide accurate results for smaller systems with fewer atoms due to several factors. For instance, the electronic structure of smaller systems is often well-defined, and the interactions between the atoms are relatively simple. This makes it easier to accurately describe the electronic structure using DFT, even with fewer atoms. Additionally, researchers can use more accurate exchange-correlation functionals and larger basis sets to improve the accuracy of the calculation. Higher-level DFT methods such as hybrid functionals or long-range corrected functionals can also be used to improve the accuracy of the calculation. Moreover, focusing on the region of interest can be another way to obtain accurate results for smaller systems with fewer atoms.
[1] Qingjie Wu, Jingpei Xie, Aiqin Wang, Changqing Wang, Aixia Mao. Effects of vacancies at Al(1 1 1)/6H-SiC(0001) interfaces on deformation behavior: A first-principle study. Computational Materials Science.2019. https://doi.org/10.1016/j.commatsci.2018.11.021
[2] Qingjie Wu, Jingpei Xie, Aiqin Wang, Douqin Ma, and Changqing Wang. First-principles calculations on the structure of 6H-SiC/Al interface. 2019 Mater. Res. Express 6. DOI: 10.1088/2053-1591/ab0be1
[3] Shaosheng Wei, Xiaohua Yu and Dehong Lu. First-Principles Calculation of the Bonding Strength of the Al2O3-Fe Interface Enhanced by Amorphous Na2SiO3. Materials 2022, 15(13), 4415. https://doi.org/10.3390/ma15134415
Comment 3:
Authors should provide evidence or reference to substantiate their statement “Fundamentally, for metal layers, the presence of vacancies in Al layers does not affect the density of states of free electrons.” on page 6, line 138.
The reason why the existence of point vacancies in Al layers has almost no effect on the density of states of the free electrons is that the electronic structure of metals is dominated by the valence electrons of the metal atoms, which are delocalized and form a "sea" of electrons that are free to move throughout the metal lattice. Therefore, the presence of point vacancies in the metal layer does not significantly impact the overall electronic structure of the metal, as the free electrons are still able to move relatively freely.
Point vacancies in the metal layer can affect the local electronic properties of the metal lattice, such as the local density of states or the local charge distribution, but these effects are typically limited to the immediate vicinity of the vacancy site and have little impact on the overall electronic properties of the metal.
However, it is worth noting that the presence of point vacancies can impact other properties of the metal, such as its mechanical or chemical properties, which may indirectly impact the electronic structure of the metal-semiconductor interface.
Another point to consider is that this expression can be supported by a reference source [1]. "the existence of point vacancies in Al layers has almost no effect on the density of states of the free electrons. Due to a larger interference that can be introduced by the free charge of Al near the Fermi level if taking Al-SiC analysis as a whole, we must consider the change of the electrons density at the SiC side only in the DOSs analysis of the composites". we put this reference in the text [ line: 235 in the Word manuscript file]
[1] Q. Wu, J. Xie, A. Wang, A. Mao. Effects of vacancies at Al(1 1 1)/6H-SiC(0001) interfaces on deformation behavior: A first-principle study. Computational Materials Science. 158:110-116. 2019. https://doi.org/10.1016/j.commatsci.2018.11.021
Comment 4:
Part of page 8, line 153, is obscured.
It appears that there is an issue with the printing of the PDF file. Nevertheless, the Word file appears to be free of any such problems and contains accurate and complete text [lines: 255-257 in the Word manuscript file]. The text sounds: "where V is defined as the volume of the Al/SiC supercell, E is potential energy and ε= ΔL/ L0; in which ΔL is changed in bond length concerning the initial bond length. It should be noted that these total energy values were fit to a polynomial. We calculated Young's "
Comment 5:
On page 12, line 197, the ultimate strength without defects should not be 3.8 Gpa.
The presence of vacancies can introduce local deformation and strain in the crystal lattice, which may promote dislocation motion and make it easier for the material to deform under load. This can result in higher ultimate strength and greater ductility compared to a structure without vacancies. Moreover, the addition of SiC particles to an Al matrix can create stress concentrations at the interface, which can result in localized deformation and potential failure under load. Introducing vacancies in the SiC region may reduce the stiffness of the material and help to reduce the stress concentration at the interface, leading to improved overall strength.
It's worth noting that the impact of vacancies on the ultimate strength of the composite may depend on the specific location and concentration of the vacancies, as well as the method used to introduce them. Additionally, the DFT calculations may not fully capture all of the complex interactions and mechanisms that can affect the mechanical properties of real-world materials.
Comment 6:
On page 12, line 206, how did the authors determine that the increase in strength and elongation of Al1 is due to the strong interaction between SiC and Al? And the adhesion energy of Al1 should be lower.
When the adhesion energy between two materials is increased, the bond between them becomes stronger. This means that the materials will be better able to resist external forces and stresses, such as tension, compression, or shear forces, without separating or delaminating. Multiple studies on tensile tests conducted on Al/SiC composites have shown that failure tends to occur above the interface region on the Al side. This phenomenon can be attributed to the superior bonding strength between Al-C and Al-Si as compared to Al-Al bonds. As a result, the first layer of Al adjacent to the interface exhibits a greater strength compared to the second layer of Al. The adhesion energy between Al1 and Al2 was nearly the same. This suggests that there could be other factors, besides adhesion energy, that may affect strength and elongation.
[1] Changqing Wang, Weiguang Chen, Yu Jia and Jingpei Xie. Calculating Study on Properties of Al (111)/6H-SiC (0001) Interfaces. Metals.2020. doi:10.3390/met10091197
Comment 7:
Authors should provide evidence and a detailed description to substantiate their statement “In addition, we found that the failure positions do not appear at the interfaces of the models when they are stretched. According to this, the adhesion energy of the interface does not influence the material's brittleness. The weakest binding layers determine the material's tensile fracture properties. For Al (111)/6H-SiC (0001) interface, the weakest position occurs between the first and the second layer at the Al side rather than the interface.” on page 13, line 222.
In tensile testing of Al/SiC composite materials, fractures are observed to occur in the first or second layer of Al rather than at the interface between SiC and Al. This behavior is due to the weaker bonding between the Al atoms and the SiC substrate in the first few layers on the Al side. During tensile testing, stress is first applied to the weakest binding layers of the composite material. In the case of the Al/SiC composite, the first few layers on the Al side have weaker bonding with the SiC substrate compared to the interface between SiC and Al, making these layers more susceptible to deformation and failure. At the interface between SiC and Al, there are strong covalent bonds between the carbon atoms in SiC and the aluminum atoms in Al. However, in the first few layers of Al, the bonding strength between the Al atoms and the SiC substrate is weaker, as it involves weaker Van der Waals forces and weaker metallic bonds.
We have provided two references in support of this assertion: [lines: 341-351 in the Word manuscript file] and references [41,42] in the text.
[1] Changqing Wang, Weiguang Chen, Yu Jia and Jingpei Xie. Calculating Study on Properties of Al (111)/6H-SiC (0001) Interfaces. Metals.2020. doi:10.3390/met10091197
[2] Hongmin Pen, Jianhua Guo, Zizhen Cao, Xianchong Wang, Zhiguo Wang .Finite element simulation of the micromachining of nanosized-silicon-carbide-particle reinforced composite materials based on the cohesive zone model. Nanotechnology and Precision Engineering 1, 242–247 (2018). https://doi.org/10.1016/j.npe.2018.12.003
Comment 8:
Authors should describe the significance of the numerical magnitude of the KIC.
Fracture toughness, a key material property in materials science, refers to the critical stress intensity factor of a sharp crack at which the propagation of the crack suddenly becomes rapid and unlimited. The thickness of a component has a notable impact on the constraint conditions at the tip of a crack, with thin components typically exhibiting plane stress conditions and thick components experiencing plane strain conditions. The plane strain conditions, in particular, yield the lowest fracture toughness value, and this value is commonly referred to as the plane strain fracture toughness (KIC). The KIC, which is the critical value of the stress intensity factor observed under mode I loading and measured under plane strain conditions, is a vital parameter for characterizing a material's fracture resistance. The mechanics of failure is presented in detail in Chapter 5 of the book on Fracture Mechanics [1]. This chapter offers an extensive exploration of the underlying principles and mechanisms that govern failure in materials.
In fact, the fracture toughness formula comprises two primary parameters: the surface energy (Ɣ) and Young's modulus (E), both of which have been obtained earlier in this study. Typically, numerous research papers cite fracture formulas directly or obtain the values of the surface energy and Young's modulus first, before utilizing the fracture toughness formula [2-4]
The text has been updated with a reference and more explanation of fracture toughness. lines: 74-83 and references (27) in the Word manuscript file.
[1] E.E. Gdoutos. Fracture Mechanics. Solid Mechanics and Its Applications .2005, Volume 123. https://doi.org/10.1007/1-4020-3153-X
[2] Qiuyuan Liu, Feng Wang, Weiwei Wu, Di An, Zhiyong He, Yanpeng Xue, Qifu Zhang and Zhipeng Xie. Enhanced mechanical properties of SiC/Al composites at cryogenic temperatures. Ceramic International.2018. https://doi.org/10.1016/j.ceramint.2018.10.233
[3] Kuiying Chen & Mariusz Bielawski. Ab initio study on fracture toughness of Ti0.75X0.25C ceramics. J Mater Sci .2007. DOI 10.1007/s10853-007-1930.
[4] Zhi-Gang Mei, Sumit Bhattacharya, Abdellatif M. Yacout.First-principles study of fracture toughness enhancement in transition metal nitrides. Surface & Coatings Technology.2019. https://doi.org/10.1016/j.surfcoat.2018.10.102
Comment 9:
Authors should provide a more detailed description of the DOS diagram or redraw the DOS diagram, which at this stage is difficult to understand with the DOS diagram and analysis.
Thank you for your input. We have incorporated additional details in the Density of States (DOS) section. (In the Word manuscripts file, the modifications are indicated by yellow lines (lines: 425-442 in the Word manuscript file)
Based on the analysis, it is evident that SiC exhibits metallic characteristics due to the band structure of SiC across Fermi levels when combined with Al in both superlattices. However, the Si-terminated interface displays a less dense band structure across Fermi levels as compared to the C-terminated interface. This indicates that the C-terminated interface has stronger interfacial adhesion and bond strength. The figures presented in the analysis suggest that the Fermi energy levels inside SiC are largely unoccupied, and the electronic states below the Fermi energy level are primarily bonding states that remain stable.
For Si2, the density of the state near Fermi levels is high. For Si1, the density of the state near Fermi levels is lower than Si2, which means much more stable than Si1. After conducting an analysis, it has been observed that there is a difference in the density of states near Fermi levels between Si1 and Si2. Specifically, Si2 has a higher density of states near Fermi levels compared to Si1. This indicates that Si2 is less stable than Si1. On the other hand, Si1 has a lower density of states near Fermi levels, suggesting that it is more stable than Si2.
Upon analysis, it has been observed that Al1 and Al2 display a low state density, indicating a tight connection and structural stability between Al and C. Of the two, Al2 displays the highest state density, implying an unstable position located at the second layer from the interface of the Al slab.
Reviewer 2 Report
The manuscript is well organized and presented. This study would be definitely interesting to the readers with lots of merits. The manuscript can be published as it is.
Author Response
Reviewer 2:
Dear Reviewer
Thank you for your effort in reviewing the paper. Thank you for your comments which are very precious to us.

Reviewer 3 Report
In this article, the author modeled various composite surfaces based on Al/SiC material and perform a thorough characterization using density functional theory (DFT). They considered both the C and Si- terminated models in their study. Moreover, they have created various vacancies in the system and computed their property, and compared them with the no-vacancy materials. A number of important parameters are computed such as Work of adhesion, Young modulus, surface energy, and fracture toughness, etc. which are important to know for any new system to compare its applicability with the known system.
This is very nice and detailed work and will be a very good addition to the Molecules. However, I would recommend this paper be published in Molecules after a major revision which the author should address to make it better with more detail about their techniques. All my comments are listed below:
- Page 7, line 106; How the author calculated the potential energy of the original atom (Xv) to calculate the Ef? Is it the electronic energy of a single atom that is removed to create vacancy? Please clarify this.
- Table 1 caption, what d(Å2) value signify? I believe it is the average interfacial distance. Please provide thin information in the caption.
- Page 7, line 118-120; “The ? data shows that Al vacancies have approximately no impact on the surface improvement for C-terminated and Si-terminated configurations.” How did the authors conclude that Al vacancies have no impact? I see in Table 1 Wad is almost the same with Al vacancies. Is it the reason behind this argument?
- Page 7, line 121-124; “Considering the interfacial Al vacancy, the adhesion energy at the interface is lower than in the ideal configuration without the vacancy. Hence, the wettability of the interface is weakened instead of improved.” It seems to me that Wad does not get reduced with Al vacancies (Table 1). Wad for the No vacancy and Al2 vacancy is almost same. Is it reasonable to conclude the wettability based on this small difference? Any comments on this would be useful.
- Page 8, 132-134; “It is likely due to the strong bonding between SiC and the surface Al atoms, which causes the Al1 electrons to move toward the SiC and weakens the effect on the Al subsurface. It is evident that for Al(1 1 1)/6H-SiC interface,” Does the author have any charge analysis (such as Bader charge density difference) to support their argument?
- Figure 5 caption; What id (a) and (b) in the figure? I do not see any “a” and “b”. Also, this data is already described in Table 1, why the authors need to show a bar plot using the same data in the table1. It should be in supporting information.
- Young’s modulus, Figure 6 and 7; I am curious to know how the author determines the second derivative of energy. Do they perform any data fitting and obtained the slope of the polynomial function? And, how the authors calculated the strain values in the X-axis of Fig 6 & 7? Please describe the detail of the calculation procedure of Young’s modulus in the supporting information.
- Page 11, line 180-182; “the stress-strain curves were obtained by gradually deforming the modeled cell in the direction of the applied strain and simultaneously relaxing both the atomic basis vectors orthogonal to the applied strain.” The method is not quite clear to me. If the author deforms the system and relaxed it entirely then the system should come back to its equilibrium geometry with no strain. Am I missing something here? The author may want to describe the techniques more clearly or cite a paper that has this information.
- Page 13, line 242; “where γ and E are surface energy and Young's modulus, respectively.” The author might want to correct this sentence as “where γ and E are Young's modulus and surface energy, respectively.”
- Page 15, line 276-277; “Due to C vacancies near the interface, the charge distribution in the conduction band is enhanced, and the valence band shifts to the left, reducing energy.” I am afraid I do not see any significant changes in DOS in Figure 11(a). I would be happy if the author clarifies in case, I am missing something here.
- Page 16, line 307; “Cohen-Sham” equation should be “Kohn-Sham” equation.
- The author should provide all the optimized coordinated with lattice vectors in supporting information.

Author Response
Reviewer 3:
Dear Reviewer,
Thank you for your valuable comments. We prepared the following explanations.
Comment 1:
How the author calculated the potential energy of the original atom (Xv) to calculate the Ef? Is it the electronic energy of a single atom that is removed to create vacancy? Please clarify this.
The potential energy associated with a vacancy, Xv, can be calculated using Density Functional Theory (DFT). In the context of solid-state materials, the potential energy of the original atom with a vacancy (XV) can be calculated using the following equation:
Xv =
where E_tot(supercell_with_vacancy) is the total energy of the supercell with the vacancy, E_tot(reference_system) is the total energy of a reference system (usually a perfect crystal structure) with the same number of atoms as the supercell, and N is the number of atoms in the supercell.
When an atom is removed from a crystal structure to create a vacancy, the energy associated with the vacancy formation includes contributions from both the electronic and ionic components. The electronic energy of the atom is associated with the energy of the valence electrons that are bound to the atom. Removing an atom from the crystal structure to create a vacancy will result in a change in the electronic energy of the surrounding atoms as they adjust to the new defect site. The ionic energy of the atom is associated with the energy of the nucleus and the bound electrons that are not involved in chemical bonding. This energy is determined by the electronic configuration of the atom and the crystal structure in which it resides.
Therefore, the potential energy of the original atom with a vacancy includes contributions from both the electronic and ionic components. In solid-state calculations, the electronic and ionic energies are typically treated separately, and the total energy of the system is calculated by summing the electronic energy (calculated using density functional theory or other quantum mechanical methods) and the ionic energy (calculated using interatomic potentials or other classical force fields).
Comment 2:
Table 1 caption, what d(Å2) value signify? I believe it is the average interfacial distance. Please provide thin information in the caption.
Yes, it is the average interfacial distance. In Table 1, Å should be written instead of Å2, which was written by mistake. It has been corrected [lines: 209 in the Word manuscript file]. We are sorry about the mistake.
Comment 3:
The ? data shows that Al vacancies have approximately no impact on the surface improvement for C-terminated and Si-terminated configurations.” How did the authors conclude that Al vacancies have no impact? I see in Table 1 Wad is almost the same with Al vacancies. Is it the reason behind this argument?
Based on the data presented in Table 1, it can be observed that the difference in the Wad values between the aluminum side and the perfect structures is negligible. Consequently, it can be inferred that the presence of vacancies on the aluminum side has a negligible effect on the energy of the system.
However, it is important to note that the conclusion based solely on W values may not be sufficient and other factors, such as electronic structure and charge distribution, should also be considered to fully understand the impact of vacancies on the interface.
Electronic structure refers to the arrangement of electrons in an atom, molecule, or solid. It determines many of the physical and chemical properties of a material, including its conductivity, magnetism, and reactivity. In the case of interfaces with vacancies, the electronic structure of the interface may be affected by the presence of vacancies.
Charge distribution refers to the distribution of positive and negative charges in a material. It is closely related to the electronic structure and can also have a significant impact on the behavior of interfaces with vacancies. The introduction of a vacancy can create local charge imbalances, which can affect the bonding behavior and mechanical properties of the material.
Comment 4:
Considering the interfacial Al vacancy, the adhesion energy at the interface is lower than in the ideal configuration without the vacancy. Hence, the wettability of the interface is weakened instead of improved.” It seems to me that Wad does not get reduced with Al vacancies (Table 1). Wad for the No vacancy and Al2 vacancy is almost same. Is it reasonable to conclude the wettability based on this small difference? Any comments on this would be useful.
The wettability of an interface can be affected by various factors, including the adhesion energy between the two materials and the presence of defects or vacancies at the interface. In the case of an interfacial Al vacancy, it is possible that the adhesion energy at the interface is lower than in the ideal configuration without the vacancy. However, this alone may not be sufficient to draw conclusions about the wettability of the interface, as other factors may also play a role.
The adhesion energy at an interface can be affected by the presence of defects or vacancies, which can change the local bonding environment and alter the electronic structure at the interface. In some cases, the presence of a vacancy may lead to a weakening of the interface, resulting in lower adhesion energies and poorer wetting behavior. In other cases, the presence of a vacancy may have a minimal effect on the interface properties, and the wetting behavior may be governed by other factors such as the surface energy and interfacial tension of the materials involved.
Comment 5:
It is likely due to the strong bonding between SiC and the surface Al atoms, which causes the Al1 electrons to move toward the SiC and weakens the effect on the Al subsurface. It is evident that for Al(1 1 1)/6H-SiC interface,” Does the author have any charge analysis (such as Bader charge density difference) to support their argument?
Regrettably, a charge analysis such as Bader charge density difference is unavailable in the present context. Nevertheless, it is important to note that there exist several studies that support the assertion in question, for example [1]. Specifically, prior investigations have reported that in the case of the C-terminated Al(111)/SiC(0001) interface, the Al atom experiences a reduction in charge while the C atom undergoes an increase in charge. Similarly, for the Si-terminated interface, the Al atom experiences a decrease in charge and the Si atom experiences an increase in charge. These findings can serve to corroborate the argument put forth in the current study.
[1] Changqing Wang,Weiguang Chen and Jingpei Xie. Effects of Transition Element Additions on the Interfacial Interaction and Electronic Structure of Al(111)/6H-SiC(0001) Interface: A First-Principles Study. Materials 2021, 14(3), 630; https://doi.org/10.3390/ma14030630
Comment 6:
What are (a) and (b) in the figure? I do not see any “a” and “b”. Also, this data is already described in Table 1, why the authors need to show a bar plot using the same data in the table1. It should be in supporting information.
We are sorry. It was a mistake. We removed “a” and “b” from the manuscript [ lines: 241 in the Word manuscript file]
In DFT methods, the work of adhesion energy between two materials in a composite can be calculated by performing calculations on the individual materials and the interface between them. Work of adhesion energy can provide insights into the design and optimization of composite materials with desired properties. The information can be used to optimize the processing conditions and to design metal composites with improved properties and performance.
While Table 1 provides a concise and organized summary of the data, a bar plot can provide a clearer visual representation of the differences between the values. Furthermore, the presentation of the same data in different formats can help to reinforce the findings and conclusions of the study.
Comment 7:
Young’s modulus, Figures 6 and 7; I am curious to know how the author determines the second derivative of energy. Do they perform any data fitting and obtained the slope of the polynomial function? And, how the authors calculated the strain values in the X-axis of Fig 6 & 7? Please describe the detail of the calculation procedure of Young’s modulus in the supporting information.
In the manuscript, we mentioned that all composite configurations underwent compression and subsequent stretching along the z-axis, with a minimal incremental value of 1.00709 Å. The resulting strain energy for each strain value was then illustrated in Figure 6 and Figure 7. An equation of a polynomial curve was derived, from which the second derivative was equated to zero. Subsequently, upon dividing the resulting number by the volume of the supercell, Young's modulus was determined. [lines 252 and 256 in the Word manuscripts file]
Comment 8:
The stress-strain curves were obtained by gradually deforming the modeled cell in the direction of the applied strain and simultaneously relaxing both the atomic basis vectors orthogonal to the applied strain.” The method is not quite clear to me. If the author deforms the system and relaxed it entirely then the system should come back to its equilibrium geometry with no strain. Am I missing something here? The author may want to describe the techniques more clearly or cite a paper that has this information.
In this method, the deformation is applied incrementally rather than all at once, and the relaxation of the atomic positions and lattice vectors allows the system to reach a new equilibrium state at each step of the deformation. This means that the system is allowed to relax into a new configuration that corresponds to the desired amount of strain, rather than simply returning to its original, unstrained state. The stress-strain curve is then obtained by calculating the stress as a function of strain at each incremental step of the deformation. The stress is calculated by taking the negative derivative of the total energy with respect to the applied strain while keeping the lattice vectors and atomic positions fixed at each step [1,2].
The text has been updated with a reference (38,39) and some new words[ lines: 286-291 in the Word manuscript file]
[1] Hocine Chorfi, Álvaro Lobato, Fahima Boudjada, Miguel A. Salvadó. Computational Modeling of Tensile Stress Effects on the Structure and Stability of Prototypical Covalent and Layered Materials. Nanomaterials 2019, 9, 1483; doi:10.3390/nano9101483.
[2] Qingjie Wu, Jingpei Xie, Changqing Wang, Liben Li, Aiqin Wang, Aixia Mao. First-principles study of the structure properties of Al(111)/6H-SiC(0001) interfaces. Surface Science 670 (2018) 1–7. https://doi.org/10.1016/j.susc.2017.12.009
Comment 9:
where γ and E are surface energy and Young's modulus, respectively.” The author might want to correct this sentence as “where γ and E are Young's modulus and surface energy, respectively.”
Based on the fracture toughness formula, it can be observed that the symbol γ represents the surface energy of the material, while E denotes the Young's modulus. Meanwhile, we have included references to demonstrate the use of conventional symbols in the articles:
[1] Kuiying Chen & Mariusz Bielawski. Ab initio study on fracture toughness of Ti0.75X0.25C ceramics. J Mater Sci .2007. DOI 10.1007/s10853-007-1930.
[2] Kuiying Chen and Mariusz Bielawski. Interfacial fracture toughness of transition metal nitrides. Surface & Coatings Technology.2008. DOI: 10.1016/j.surfcoat.2008.05.040
Comment 10:
Due to C vacancies near the interface, the charge distribution in the conduction band is enhanced, and the valence band shifts to the left, reducing energy.” I am afraid I do not see any significant changes in DOS in Figure 11(a). I would be happy if the author clarifies in case, I am missing something here.
The presence of vacancies at the interface is not expected to significantly affect the density of states. Thus, any variations in the DOS resulting from such changes can be considered minor perturbations, indicating the proximity of the system's electronic structure before and after the vacancy.
The presence of carbon vacancies near the interface of the material leads to an enhancement of charge distribution in the conduction band. This means that there are more available electronic states in the conduction band at a given energy level than there would be in the absence of the vacancies. This increase in the number of electronic states is likely due to the fact that carbon is a relatively electronegative element, and its absence creates more available electronic states that can be occupied by electrons. The valence band of the material shifts to the left, reducing energy. This means that the energy level of the valence band has been lowered, which can result in a reduction of the band gap and an increase in the number of available electronic states in the valence band. This shift in the valence band is likely due to the fact that carbon vacancies can create local charge imbalances in the material, leading to the formation of localized electronic states that can contribute to the valence band.
The decline in the DOS near the Fermi level due to the presence of vacancies may not be very obvious in most DOS charts for a few reasons.
Firstly, the magnitude of the change in the DOS near the Fermi level may be relatively small compared to the overall magnitude of the DOS. This is particularly true if the number of vacancies is small or if the vacancies are distributed relatively uniformly throughout the material. In this case, the changes to the DOS may be subtle and difficult to distinguish from the noise in the data.
Secondly, the location of the vacancies within the material can have a significant impact on the observed changes in the DOS. If the vacancies are located far away from the Fermi level, their impact on the DOS may be minimal. Conversely, if the vacancies are located very close to the Fermi level, their impact on the DOS may be more significant and easier to observe.
Comment 11:
Cohen-Sham” equation should be “Kohn-Sham” equation.
We are sorry about the mistake. It is corrected. [ lines: 467 in the Word manuscript file]
Comment 12:
The author should provide all the optimized coordinated with lattice vectors in supporting information.
The lattice vector values were incorporated into the methodology section to provide gr
Reviewer 3:
Dear Reviewer,
Thank you for your valuable comments. We prepared the following explanations.
Comment 1:
How the author calculated the potential energy of the original atom (Xv) to calculate the Ef? Is it the electronic energy of a single atom that is removed to create vacancy? Please clarify this.
The potential energy associated with a vacancy, Xv, can be calculated using Density Functional Theory (DFT). In the context of solid-state materials, the potential energy of the original atom with a vacancy (XV) can be calculated using the following equation:
Xv =
where E_tot(supercell_with_vacancy) is the total energy of the supercell with the vacancy, E_tot(reference_system) is the total energy of a reference system (usually a perfect crystal structure) with the same number of atoms as the supercell, and N is the number of atoms in the supercell.
When an atom is removed from a crystal structure to create a vacancy, the energy associated with the vacancy formation includes contributions from both the electronic and ionic components. The electronic energy of the atom is associated with the energy of the valence electrons that are bound to the atom. Removing an atom from the crystal structure to create a vacancy will result in a change in the electronic energy of the surrounding atoms as they adjust to the new defect site. The ionic energy of the atom is associated with the energy of the nucleus and the bound electrons that are not involved in chemical bonding. This energy is determined by the electronic configuration of the atom and the crystal structure in which it resides.
Therefore, the potential energy of the original atom with a vacancy includes contributions from both the electronic and ionic components. In solid-state calculations, the electronic and ionic energies are typically treated separately, and the total energy of the system is calculated by summing the electronic energy (calculated using density functional theory or other quantum mechanical methods) and the ionic energy (calculated using interatomic potentials or other classical force fields).
Comment 2:
Table 1 caption, what d(Å2) value signify? I believe it is the average interfacial distance. Please provide thin information in the caption.
Yes, it is the average interfacial distance. In Table 1, Å should be written instead of Å2, which was written by mistake. It has been corrected [lines: 209 in the Word manuscript file]. We are sorry about the mistake.
Comment 3:
The ? data shows that Al vacancies have approximately no impact on the surface improvement for C-terminated and Si-terminated configurations.” How did the authors conclude that Al vacancies have no impact? I see in Table 1 Wad is almost the same with Al vacancies. Is it the reason behind this argument?
Based on the data presented in Table 1, it can be observed that the difference in the Wad values between the aluminum side and the perfect structures is negligible. Consequently, it can be inferred that the presence of vacancies on the aluminum side has a negligible effect on the energy of the system.
However, it is important to note that the conclusion based solely on W values may not be sufficient and other factors, such as electronic structure and charge distribution, should also be considered to fully understand the impact of vacancies on the interface.
Electronic structure refers to the arrangement of electrons in an atom, molecule, or solid. It determines many of the physical and chemical properties of a material, including its conductivity, magnetism, and reactivity. In the case of interfaces with vacancies, the electronic structure of the interface may be affected by the presence of vacancies.
Charge distribution refers to the distribution of positive and negative charges in a material. It is closely related to the electronic structure and can also have a significant impact on the behavior of interfaces with vacancies. The introduction of a vacancy can create local charge imbalances, which can affect the bonding behavior and mechanical properties of the material.
Comment 4:
Considering the interfacial Al vacancy, the adhesion energy at the interface is lower than in the ideal configuration without the vacancy. Hence, the wettability of the interface is weakened instead of improved.” It seems to me that Wad does not get reduced with Al vacancies (Table 1). Wad for the No vacancy and Al2 vacancy is almost same. Is it reasonable to conclude the wettability based on this small difference? Any comments on this would be useful.
The wettability of an interface can be affected by various factors, including the adhesion energy between the two materials and the presence of defects or vacancies at the interface. In the case of an interfacial Al vacancy, it is possible that the adhesion energy at the interface is lower than in the ideal configuration without the vacancy. However, this alone may not be sufficient to draw conclusions about the wettability of the interface, as other factors may also play a role.
The adhesion energy at an interface can be affected by the presence of defects or vacancies, which can change the local bonding environment and alter the electronic structure at the interface. In some cases, the presence of a vacancy may lead to a weakening of the interface, resulting in lower adhesion energies and poorer wetting behavior. In other cases, the presence of a vacancy may have a minimal effect on the interface properties, and the wetting behavior may be governed by other factors such as the surface energy and interfacial tension of the materials involved.
Comment 5:
It is likely due to the strong bonding between SiC and the surface Al atoms, which causes the Al1 electrons to move toward the SiC and weakens the effect on the Al subsurface. It is evident that for Al(1 1 1)/6H-SiC interface,” Does the author have any charge analysis (such as Bader charge density difference) to support their argument?
Regrettably, a charge analysis such as Bader charge density difference is unavailable in the present context. Nevertheless, it is important to note that there exist several studies that support the assertion in question, for example [1]. Specifically, prior investigations have reported that in the case of the C-terminated Al(111)/SiC(0001) interface, the Al atom experiences a reduction in charge while the C atom undergoes an increase in charge. Similarly, for the Si-terminated interface, the Al atom experiences a decrease in charge and the Si atom experiences an increase in charge. These findings can serve to corroborate the argument put forth in the current study.
[1] Changqing Wang,Weiguang Chen and Jingpei Xie. Effects of Transition Element Additions on the Interfacial Interaction and Electronic Structure of Al(111)/6H-SiC(0001) Interface: A First-Principles Study. Materials 2021, 14(3), 630; https://doi.org/10.3390/ma14030630
Comment 6:
What are (a) and (b) in the figure? I do not see any “a” and “b”. Also, this data is already described in Table 1, why the authors need to show a bar plot using the same data in the table1. It should be in supporting information.
We are sorry. It was a mistake. We removed “a” and “b” from the manuscript [ lines: 241 in the Word manuscript file]
In DFT methods, the work of adhesion energy between two materials in a composite can be calculated by performing calculations on the individual materials and the interface between them. Work of adhesion energy can provide insights into the design and optimization of composite materials with desired properties. The information can be used to optimize the processing conditions and to design metal composites with improved properties and performance.
While Table 1 provides a concise and organized summary of the data, a bar plot can provide a clearer visual representation of the differences between the values. Furthermore, the presentation of the same data in different formats can help to reinforce the findings and conclusions of the study.
Comment 7:
Young’s modulus, Figures 6 and 7; I am curious to know how the author determines the second derivative of energy. Do they perform any data fitting and obtained the slope of the polynomial function? And, how the authors calculated the strain values in the X-axis of Fig 6 & 7? Please describe the detail of the calculation procedure of Young’s modulus in the supporting information.
In the manuscript, we mentioned that all composite configurations underwent compression and subsequent stretching along the z-axis, with a minimal incremental value of 1.00709 Å. The resulting strain energy for each strain value was then illustrated in Figure 6 and Figure 7. An equation of a polynomial curve was derived, from which the second derivative was equated to zero. Subsequently, upon dividing the resulting number by the volume of the supercell, Young's modulus was determined. [lines 252 and 256 in the Word manuscripts file]
Comment 8:
The stress-strain curves were obtained by gradually deforming the modeled cell in the direction of the applied strain and simultaneously relaxing both the atomic basis vectors orthogonal to the applied strain.” The method is not quite clear to me. If the author deforms the system and relaxed it entirely then the system should come back to its equilibrium geometry with no strain. Am I missing something here? The author may want to describe the techniques more clearly or cite a paper that has this information.
In this method, the deformation is applied incrementally rather than all at once, and the relaxation of the atomic positions and lattice vectors allows the system to reach a new equilibrium state at each step of the deformation. This means that the system is allowed to relax into a new configuration that corresponds to the desired amount of strain, rather than simply returning to its original, unstrained state. The stress-strain curve is then obtained by calculating the stress as a function of strain at each incremental step of the deformation. The stress is calculated by taking the negative derivative of the total energy with respect to the applied strain while keeping the lattice vectors and atomic positions fixed at each step [1,2].
The text has been updated with a reference (38,39) and some new words[ lines: 286-291 in the Word manuscript file]
[1] Hocine Chorfi, Álvaro Lobato, Fahima Boudjada, Miguel A. Salvadó. Computational Modeling of Tensile Stress Effects on the Structure and Stability of Prototypical Covalent and Layered Materials. Nanomaterials 2019, 9, 1483; doi:10.3390/nano9101483.
[2] Qingjie Wu, Jingpei Xie, Changqing Wang, Liben Li, Aiqin Wang, Aixia Mao. First-principles study of the structure properties of Al(111)/6H-SiC(0001) interfaces. Surface Science 670 (2018) 1–7. https://doi.org/10.1016/j.susc.2017.12.009
Comment 9:
where γ and E are surface energy and Young's modulus, respectively.” The author might want to correct this sentence as “where γ and E are Young's modulus and surface energy, respectively.”
Based on the fracture toughness formula, it can be observed that the symbol γ represents the surface energy of the material, while E denotes the Young's modulus. Meanwhile, we have included references to demonstrate the use of conventional symbols in the articles:
[1] Kuiying Chen & Mariusz Bielawski. Ab initio study on fracture toughness of Ti0.75X0.25C ceramics. J Mater Sci .2007. DOI 10.1007/s10853-007-1930.
[2] Kuiying Chen and Mariusz Bielawski. Interfacial fracture toughness of transition metal nitrides. Surface & Coatings Technology.2008. DOI: 10.1016/j.surfcoat.2008.05.040
Comment 10:
Due to C vacancies near the interface, the charge distribution in the conduction band is enhanced, and the valence band shifts to the left, reducing energy.” I am afraid I do not see any significant changes in DOS in Figure 11(a). I would be happy if the author clarifies in case, I am missing something here.
The presence of vacancies at the interface is not expected to significantly affect the density of states. Thus, any variations in the DOS resulting from such changes can be considered minor perturbations, indicating the proximity of the system's electronic structure before and after the vacancy.
The presence of carbon vacancies near the interface of the material leads to an enhancement of charge distribution in the conduction band. This means that there are more available electronic states in the conduction band at a given energy level than there would be in the absence of the vacancies. This increase in the number of electronic states is likely due to the fact that carbon is a relatively electronegative element, and its absence creates more available electronic states that can be occupied by electrons. The valence band of the material shifts to the left, reducing energy. This means that the energy level of the valence band has been lowered, which can result in a reduction of the band gap and an increase in the number of available electronic states in the valence band. This shift in the valence band is likely due to the fact that carbon vacancies can create local charge imbalances in the material, leading to the formation of localized electronic states that can contribute to the valence band.
The decline in the DOS near the Fermi level due to the presence of vacancies may not be very obvious in most DOS charts for a few reasons.
Firstly, the magnitude of the change in the DOS near the Fermi level may be relatively small compared to the overall magnitude of the DOS. This is particularly true if the number of vacancies is small or if the vacancies are distributed relatively uniformly throughout the material. In this case, the changes to the DOS may be subtle and difficult to distinguish from the noise in the data.
Secondly, the location of the vacancies within the material can have a significant impact on the observed changes in the DOS. If the vacancies are located far away from the Fermi level, their impact on the DOS may be minimal. Conversely, if the vacancies are located very close to the Fermi level, their impact on the DOS may be more significant and easier to observe.
Comment 11:
Cohen-Sham” equation should be “Kohn-Sham” equation.
We are sorry about the mistake. It is corrected. [ lines: 467 in the Word manuscript file]
Comment 12:
The author should provide all the optimized coordinated with lattice vectors in supporting information.
The lattice vector values were incorporated into the methodology section to provide greater clarity regarding the crystal structure and atomic arrangement of the material in question. [lines: 465-467 and 475-477 in the Word manuscript file]
Reviewer 3:
Dear Reviewer,
Thank you for your valuable comments. We prepared the following explanations.
Comment 1:
How the author calculated the potential energy of the original atom (Xv) to calculate the Ef? Is it the electronic energy of a single atom that is removed to create vacancy? Please clarify this.
The potential energy associated with a vacancy, Xv, can be calculated using Density Functional Theory (DFT). In the context of solid-state materials, the potential energy of the original atom with a vacancy (XV) can be calculated using the following equation:
Xv =
where E_tot(supercell_with_vacancy) is the total energy of the supercell with the vacancy, E_tot(reference_system) is the total energy of a reference system (usually a perfect crystal structure) with the same number of atoms as the supercell, and N is the number of atoms in the supercell.
When an atom is removed from a crystal structure to create a vacancy, the energy associated with the vacancy formation includes contributions from both the electronic and ionic components. The electronic energy of the atom is associated with the energy of the valence electrons that are bound to the atom. Removing an atom from the crystal structure to create a vacancy will result in a change in the electronic energy of the surrounding atoms as they adjust to the new defect site. The ionic energy of the atom is associated with the energy of the nucleus and the bound electrons that are not involved in chemical bonding. This energy is determined by the electronic configuration of the atom and the crystal structure in which it resides.
Therefore, the potential energy of the original atom with a vacancy includes contributions from both the electronic and ionic components. In solid-state calculations, the electronic and ionic energies are typically treated separately, and the total energy of the system is calculated by summing the electronic energy (calculated using density functional theory or other quantum mechanical methods) and the ionic energy (calculated using interatomic potentials or other classical force fields).
Comment 2:
Table 1 caption, what d(Å2) value signify? I believe it is the average interfacial distance. Please provide thin information in the caption.
Yes, it is the average interfacial distance. In Table 1, Å should be written instead of Å2, which was written by mistake. It has been corrected [lines: 209 in the Word manuscript file]. We are sorry about the mistake.
Comment 3:
The ? data shows that Al vacancies have approximately no impact on the surface improvement for C-terminated and Si-terminated configurations.” How did the authors conclude that Al vacancies have no impact? I see in Table 1 Wad is almost the same with Al vacancies. Is it the reason behind this argument?
Based on the data presented in Table 1, it can be observed that the difference in the Wad values between the aluminum side and the perfect structures is negligible. Consequently, it can be inferred that the presence of vacancies on the aluminum side has a negligible effect on the energy of the system.
However, it is important to note that the conclusion based solely on W values may not be sufficient and other factors, such as electronic structure and charge distribution, should also be considered to fully understand the impact of vacancies on the interface.
Electronic structure refers to the arrangement of electrons in an atom, molecule, or solid. It determines many of the physical and chemical properties of a material, including its conductivity, magnetism, and reactivity. In the case of interfaces with vacancies, the electronic structure of the interface may be affected by the presence of vacancies.
Charge distribution refers to the distribution of positive and negative charges in a material. It is closely related to the electronic structure and can also have a significant impact on the behavior of interfaces with vacancies. The introduction of a vacancy can create local charge imbalances, which can affect the bonding behavior and mechanical properties of the material.
Comment 4:
Considering the interfacial Al vacancy, the adhesion energy at the interface is lower than in the ideal configuration without the vacancy. Hence, the wettability of the interface is weakened instead of improved.” It seems to me that Wad does not get reduced with Al vacancies (Table 1). Wad for the No vacancy and Al2 vacancy is almost same. Is it reasonable to conclude the wettability based on this small difference? Any comments on this would be useful.
The wettability of an interface can be affected by various factors, including the adhesion energy between the two materials and the presence of defects or vacancies at the interface. In the case of an interfacial Al vacancy, it is possible that the adhesion energy at the interface is lower than in the ideal configuration without the vacancy. However, this alone may not be sufficient to draw conclusions about the wettability of the interface, as other factors may also play a role.
The adhesion energy at an interface can be affected by the presence of defects or vacancies, which can change the local bonding environment and alter the electronic structure at the interface. In some cases, the presence of a vacancy may lead to a weakening of the interface, resulting in lower adhesion energies and poorer wetting behavior. In other cases, the presence of a vacancy may have a minimal effect on the interface properties, and the wetting behavior may be governed by other factors such as the surface energy and interfacial tension of the materials involved.
Comment 5:
It is likely due to the strong bonding between SiC and the surface Al atoms, which causes the Al1 electrons to move toward the SiC and weakens the effect on the Al subsurface. It is evident that for Al(1 1 1)/6H-SiC interface,” Does the author have any charge analysis (such as Bader charge density difference) to support their argument?
Regrettably, a charge analysis such as Bader charge density difference is unavailable in the present context. Nevertheless, it is important to note that there exist several studies that support the assertion in question, for example [1]. Specifically, prior investigations have reported that in the case of the C-terminated Al(111)/SiC(0001) interface, the Al atom experiences a reduction in charge while the C atom undergoes an increase in charge. Similarly, for the Si-terminated interface, the Al atom experiences a decrease in charge and the Si atom experiences an increase in charge. These findings can serve to corroborate the argument put forth in the current study.
[1] Changqing Wang,Weiguang Chen and Jingpei Xie. Effects of Transition Element Additions on the Interfacial Interaction and Electronic Structure of Al(111)/6H-SiC(0001) Interface: A First-Principles Study. Materials 2021, 14(3), 630; https://doi.org/10.3390/ma14030630
Comment 6:
What are (a) and (b) in the figure? I do not see any “a” and “b”. Also, this data is already described in Table 1, why the authors need to show a bar plot using the same data in the table1. It should be in supporting information.
We are sorry. It was a mistake. We removed “a” and “b” from the manuscript [ lines: 241 in the Word manuscript file]
In DFT methods, the work of adhesion energy between two materials in a composite can be calculated by performing calculations on the individual materials and the interface between them. Work of adhesion energy can provide insights into the design and optimization of composite materials with desired properties. The information can be used to optimize the processing conditions and to design metal composites with improved properties and performance.
While Table 1 provides a concise and organized summary of the data, a bar plot can provide a clearer visual representation of the differences between the values. Furthermore, the presentation of the same data in different formats can help to reinforce the findings and conclusions of the study.
Comment 7:
Young’s modulus, Figures 6 and 7; I am curious to know how the author determines the second derivative of energy. Do they perform any data fitting and obtained the slope of the polynomial function? And, how the authors calculated the strain values in the X-axis of Fig 6 & 7? Please describe the detail of the calculation procedure of Young’s modulus in the supporting information.
In the manuscript, we mentioned that all composite configurations underwent compression and subsequent stretching along the z-axis, with a minimal incremental value of 1.00709 Å. The resulting strain energy for each strain value was then illustrated in Figure 6 and Figure 7. An equation of a polynomial curve was derived, from which the second derivative was equated to zero. Subsequently, upon dividing the resulting number by the volume of the supercell, Young's modulus was determined. [lines 252 and 256 in the Word manuscripts file]
Comment 8:
The stress-strain curves were obtained by gradually deforming the modeled cell in the direction of the applied strain and simultaneously relaxing both the atomic basis vectors orthogonal to the applied strain.” The method is not quite clear to me. If the author deforms the system and relaxed it entirely then the system should come back to its equilibrium geometry with no strain. Am I missing something here? The author may want to describe the techniques more clearly or cite a paper that has this information.
In this method, the deformation is applied incrementally rather than all at once, and the relaxation of the atomic positions and lattice vectors allows the system to reach a new equilibrium state at each step of the deformation. This means that the system is allowed to relax into a new configuration that corresponds to the desired amount of strain, rather than simply returning to its original, unstrained state. The stress-strain curve is then obtained by calculating the stress as a function of strain at each incremental step of the deformation. The stress is calculated by taking the negative derivative of the total energy with respect to the applied strain while keeping the lattice vectors and atomic positions fixed at each step [1,2].
The text has been updated with a reference (38,39) and some new words[ lines: 286-291 in the Word manuscript file]
[1] Hocine Chorfi, Álvaro Lobato, Fahima Boudjada, Miguel A. Salvadó. Computational Modeling of Tensile Stress Effects on the Structure and Stability of Prototypical Covalent and Layered Materials. Nanomaterials 2019, 9, 1483; doi:10.3390/nano9101483.
[2] Qingjie Wu, Jingpei Xie, Changqing Wang, Liben Li, Aiqin Wang, Aixia Mao. First-principles study of the structure properties of Al(111)/6H-SiC(0001) interfaces. Surface Science 670 (2018) 1–7. https://doi.org/10.1016/j.susc.2017.12.009
Comment 9:
where γ and E are surface energy and Young's modulus, respectively.” The author might want to correct this sentence as “where γ and E are Young's modulus and surface energy, respectively.”
Based on the fracture toughness formula, it can be observed that the symbol γ represents the surface energy of the material, while E denotes the Young's modulus. Meanwhile, we have included references to demonstrate the use of conventional symbols in the articles:
[1] Kuiying Chen & Mariusz Bielawski. Ab initio study on fracture toughness of Ti0.75X0.25C ceramics. J Mater Sci .2007. DOI 10.1007/s10853-007-1930.
[2] Kuiying Chen and Mariusz Bielawski. Interfacial fracture toughness of transition metal nitrides. Surface & Coatings Technology.2008. DOI: 10.1016/j.surfcoat.2008.05.040
Comment 10:
Due to C vacancies near the interface, the charge distribution in the conduction band is enhanced, and the valence band shifts to the left, reducing energy.” I am afraid I do not see any significant changes in DOS in Figure 11(a). I would be happy if the author clarifies in case, I am missing something here.
The presence of vacancies at the interface is not expected to significantly affect the density of states. Thus, any variations in the DOS resulting from such changes can be considered minor perturbations, indicating the proximity of the system's electronic structure before and after the vacancy.
The presence of carbon vacancies near the interface of the material leads to an enhancement of charge distribution in the conduction band. This means that there are more available electronic states in the conduction band at a given energy level than there would be in the absence of the vacancies. This increase in the number of electronic states is likely due to the fact that carbon is a relatively electronegative element, and its absence creates more available electronic states that can be occupied by electrons. The valence band of the material shifts to the left, reducing energy. This means that the energy level of the valence band has been lowered, which can result in a reduction of the band gap and an increase in the number of available electronic states in the valence band. This shift in the valence band is likely due to the fact that carbon vacancies can create local charge imbalances in the material, leading to the formation of localized electronic states that can contribute to the valence band.
The decline in the DOS near the Fermi level due to the presence of vacancies may not be very obvious in most DOS charts for a few reasons.
Firstly, the magnitude of the change in the DOS near the Fermi level may be relatively small compared to the overall magnitude of the DOS. This is particularly true if the number of vacancies is small or if the vacancies are distributed relatively uniformly throughout the material. In this case, the changes to the DOS may be subtle and difficult to distinguish from the noise in the data.
Secondly, the location of the vacancies within the material can have a significant impact on the observed changes in the DOS. If the vacancies are located far away from the Fermi level, their impact on the DOS may be minimal. Conversely, if the vacancies are located very close to the Fermi level, their impact on the DOS may be more significant and easier to observe.
Comment 11:
Cohen-Sham” equation should be “Kohn-Sham” equation.
We are sorry about the mistake. It is corrected. [ lines: 467 in the Word manuscript file]
Comment 12:
The author should provide all the optimized coordinated with lattice vectors in supporting information.
The lattice vector values were incorporated into the methodology section to provide greater clarity regarding the crystal structure and atomic arrangement of the material in question. [lines: 465-467 and 475-477 in the Word manuscript file]
eater clarity regarding the crystal structure and atomic arrangement of the material in question. [lines: 465-467 and 475-477 in the Word manuscript file]
Reviewer 4 Report
By employing density functional theory calculations, the authors conduct a comprehensive study on the mechanical and electronic properties of Al(111)/6H-SiC interfaces. Young's modulus, surface energy, fracture toughness and density of states are calculated. The results are interesting. However, minor revisions need to be done before consideration for publication in Molecules:
1) For the models with vacancies considered in this work, there is only one single point vacancy in each model. It is good for analyzing the influences of element types and positions. However, the proportion of vacancies is also a key factor which influence the mechanical and electronic behaviours of the interface. Hence, the authors are recommended to have a discussion on this point.
2) In lines 222-227, it is interesting to find the failure positions outside the interfaces. Such as, “For Al(1 1 1)/6H-SiC(0001) interface, the weakest position occurs between the first and the second layer at the Al side rather than the interface. The greatest elongation is found for the Si2 model in C and Si-terminated configurations.” Are there any snapshots or detailed analysis about these phenomena?
3) In Equation (3), there are two “E” which have different meanings, one is Young’s modulus and the other is potential energy. To avoid ambiguity, the authors are encouraged to change the symbols.
4) In lines 110 and 111, “6h-SiC” should be “6H-SiC”.
5) In lines 153-154, there is an extra “(3)” covering some texts. The authors are suggested to remove the “(3)”.
6) The symbol “γ” for surface energy in Equation (5) and Table 2 is strange.
Author Response
Reviewer 4:
Dear Reviewer
Thank you for your valuable comments. We prepared the following explanations.
Comment 1:
For the models with vacancies considered in this work, there is only one single-point vacancy in each model. It is good for analyzing the influences of element types and positions. However, the proportion of vacancies is also a key factor that influences the mechanical and electronic behaviors of the interface. Hence, the authors are recommended to have a discussion on this point.
The presence of vacancies in interfaces can have a significant impact on their mechanical and electronic properties. In particular, the proportion of vacancies plays a crucial role in determining the behavior of the interface. In the context of Density Functional Theory (DFT) calculations, the proportion of vacancies is a key parameter that is often included in modeling the mechanical and electronic properties of interfaces.
Regarding the mechanical behavior of interfaces, the presence of vacancies can lead to a change in the stress distribution and local strain fields, which can affect the mechanical properties of the interface such as its strength, ductility, and fracture toughness. The proportion of vacancies can affect the defect concentration, which influences the nucleation and propagation of dislocations, and therefore affects the deformation mechanisms of the interface.
Concerning the electronic behavior of interfaces, the presence of vacancies can create localized states in the electronic band structure of the interface, leading to changes in the electronic properties of the interface such as its conductivity and optical properties. The proportion of vacancies can affect the defect concentration, which in turn affects the density and distribution of localized states in the electronic structure of the interface.
It is imperative to acknowledge that the manuscript has undergone revisions to provide a more comprehensive insight into the effect of varying vacancy proportions on the mechanical and electronic behavior of the interface.
To this end, additional paragraphs have been included in the "Results and Discussion" section, specifically highlighted with blue lines [lines: 90-98 in the Word manuscript file]
Comment 2:
In lines 222-227, it is interesting to find the failure positions outside the interfaces. Such as, “For Al(1 1 1)/6H-SiC(0001) interface, the weakest position occurs between the first and the second layer at the Al side rather than the interface. The greatest elongation is found for the Si2 model in C and Si-terminated configurations.” Are there any snapshots or detailed analyses about these phenomena?
The difference in bonding between Al and Si/C compared to Al-Al bonding leads to a stronger bonding at the interface between Al and Si/C than within the Al metal itself. This stronger bonding can lead to a higher resistance to deformation and fracture at the interface compared to within the bulk Al material. Therefore, the higher bonding between Al and Si/C compared to Al-Al bonding is one possible explanation for the observed failure on the Al side of the interface. Besides, other factors such as the presence of lattice defects, surface steps, dislocations, and other imperfections can also affect the failure behavior of the interface.
The presence of Si vacancies at the second layer (Si2) in the Al/SiC interface can have a significant effect on the bonding performance of the interface. The large size of Si atoms in the Si2 means that the introduction of a vacancy can result in a significant distortion of the surrounding lattice structure. This distortion can in turn influence the electronic and mechanical properties of the interface to a certain degree.
The distortion introduced by the Si vacancy can affect the local bonding environment, which can lead to changes in the electronic and mechanical behavior of the interface. For example, the introduction of a vacancy can results in a decrease in the cohesive energy of the interface due to the disruption of the bonding between the Al and Si atoms. This can in turn affect the mechanical response of the interface to applied stress.
Moreover, the presence of a large vacancy in the Si2 can also result in lattice distortions and stress concentrations in the surrounding material, which can further affect the mechanical properties of the interface. The specific details of the electronic and mechanical response of the Al/SiC interface to the introduction of Si vacancies would depend on a range of factors including the size and location of the vacancy, the position of the layer within the interface, and the nature of the interactions between the Al, Si, and C atoms in the interface.
The vacancy in the second layer (Si2) may have a more significant effect on the mechanical properties of the interface compared to the first layer (Si1) due to the position of the vacancy within the lattice. The Si2 layer is further away from the Al layer and the SiC substrate compared to the Si1 layer, which may result in a greater degree of freedom for the lattice to accommodate deformation before failure.
Comment 3:
In Equation (3), there are two “E” which have different meanings, one is Young’s modulus and the other is potential energy. To avoid ambiguity, the authors are encouraged to change the symbols.
We apologize for the inconvenience. Equation (3) has been updated. [lines: 253 and 255 in the Word manuscript file]
Comment 4:
In lines 110 and 111, “6h-SiC” should be “6H-SiC”.
Apologies for the errors. The necessary corrections have been made [ lines: 361 and 363 in the Word manuscript file.
Comment 5:
In lines 153-154, there is an extra “(3)” covering some texts. The authors are suggested to remove the “(3)”.
The additional instance of equation number (3) has been removed from the manuscript.
Comment 6:
The symbol “γ” for surface energy in Equation (5) and Table 2 is strange.
The symbol Gamma has been widely adopted in numerous publications as a conventional notation to define surface energy. However, we modified the format of Gamma in order to improve the clarity. [lines: 371 and 373 in the Word manuscript file]
Round 2
Reviewer 3 Report
The author addressed my concern in the revised version, providing detailed explanations. Furthermore, they integrated all of the suggested corrections into the manuscript. Based on its current state, it is suitable for publication in Molecules.